# Facial Synthesis and Bioevaluation of Well-Defined OEGylated Betulinic Acid-Cyclodextrin Conjugates for Inhibition of Influenza Infection

**DOI:** 10.3390/molecules27041163

**Published:** 2022-02-09

**Authors:** Yingying Chen, Xinchen Wang, Xinyuan Ma, Shuobin Liang, Qianqian Gao, Elena V. Tretyakova, Yongmin Zhang, Demin Zhou, Sulong Xiao

**Affiliations:** 1State Key Laboratory of Natural and Biomimetic Drugs, School of Pharmaceutical Sciences, Peking University, Beijing 100191, China; chenyy@bjmu.edu.cn (Y.C.); xinchen_w@163.com (X.W.); 2011210051@bjmu.edu.cn (X.M.); binl9508@163.com (S.L.); 15104675690@163.com (Q.G.); deminzhou@bjmu.edu.cn (D.Z.); 2Ufa Institute of Chemistry of the Ufa Federal Research Centre of the Russian Academy of Sciences, 71 Prospect Oktyabrya, 450054 Ufa, Russia; tretyakovaelv@gmail.com; 3Sorbonne Université, CNRS, Institut Parisien de Chimie Moléculaire, UMR 8232, 4 Place Jussieu, 75005 Paris, France; yongmin.zhang@upmc.fr; 4Shenzhen Bay Laboratory, Institute of Chemical Biology, Shenzhen 518132, China

**Keywords:** lupane triterpene, click chemistry, antiviral activity, structure-activity relationships, multivalent

## Abstract

Betulinic acid (BA) and its derivatives exhibit a variety of biological activities, especially their anti-HIV-1 activity, but generally have only modest inhibitory potency against influenza virus. The entry of influenza virus into host cells can be competitively inhibited by multivalent derivatives targeting hemagglutinin. In this study, a series of hexa-, hepta- and octavalent BA derivatives based on α-, β- and γ-cyclodextrin scaffolds, respectively, with varying lengths of flexible oligo(ethylene glycol) linkers was designed and synthesized using a microwave-assisted copper-catalyzed 1,3-dipolar cycloaddition reaction. The generated BA-cyclodextrin conjugates were tested for their in vitro activity against influenza A/WSN/33 (H1N1) virus and cytotoxicity. Among the tested compounds, **58**, **80** and **82** showed slight cytotoxicity to Madin-Darby canine kidney cells with viabilities ranging from 64 to 68% at a high concentration of 100 μM. Four conjugates **51** and **69**–**71** showed significant inhibitory effects on influenza infection with half maximal inhibitory concentration values of 5.20, 9.82, 7.48 and 7.59 μM, respectively. The structure-activity relationships of multivalent BA-cyclodextrin conjugates were discussed, highlighting that multivalent BA derivatives may be potential antiviral agents against influenza infection.

## 1. Introduction

Influenza viruses are widespread human respiratory pathogens that can cause serious infections with significant morbidity and mortality [1]. Recently, coinfection of influenza virus with severe acute respiratory syndrome coronavirus 2 (SARS-CoV-2) has been reported [2], highlighting that the prevention and treatment of influenza will be more important than ever. Due to the lack of activity against influenza B and the widespread resistance of M2 ion channel inhibitors among circulating influenza strains, the antiviral drugs currently recommended for the treatment of influenza are limited to neuraminidase (NA) [3] and polymerase acidic protein (PA) inhibitors [4]. Although variants resistant to NA and PA inhibitors are much less than M2 inhibitors, the high variability of influenza viruses, such as seasonal H1N1 viruses carrying the H275Y, H275Y and I38T mutations [5] and the H7N9 virus carrying the R294K mutation [6], enables the rapid evolution of antiviral resistance to drugs, underscoring the urgent need for the development of new anti-influenza drugs.

The entry of influenza virus into host cells is a six-step dynamic process [7], which represents an attractive target for antiviral therapy. Influenza viruses attach to host cells by binding the globular head of hemagglutinin (HA), a homotrimeric type I membrane glycoprotein expressed on the virion surface, to sialylated host cells. The interaction between HA and sialic acid is usually weak with an association constant of 10^3^ M^−^^1^ [8]. However, the interactions between multiple HA trimers on the viral surface (~600–1200 molecules per virus particle) and sialic acid-terminated glycoproteins and glycolipids on the cell surface (~50–200 residues per 100 nm^2^) substantially increase through multivalent effects [9]. To this end, linear polymers [10], dendritic polymers [11], and nanoparticles [12] have been used as different display systems for highly potent influenza virus inhibitors. We have previously reported the synthesis of multivalent pentacyclic triterpene conjugates [13] and found that three heptavalent pentacyclic triterpene derivatives **1**–**3** display broad-spectrum anti-influenza virus activity with half maximal inhibitory concentration (IC_50_) values in the 1.60–18.74 μM range (Figure 1). Two years later, Li et al. demonstrated the synthesis of a series of random glycyrrhetinic acid/oligo(ethylene glycol) (OEG)-appended norbomene copolymers **4** as potential nanocarriers for drug delivery [14]. In another study, Yang et al. reported the synthesis of PEGylated oleanolic acid-functionalized human serum albumin conjugates **5**–**7** and their potential use as anti-infective agents [15]. These results rationalize the construction of multivalent pentacyclic triterpenes as potential inhibitors to block the replication of influenza viruses.

Betulinic acid (BA) is a naturally occurring pentacyclic triterpenoid found in several species of plants, notably *Betula pubescens*, commonly known as white birch. Owing to its unusual multiple biological effects, BA has garnered attention from researchers in the scientific community and pharmaceutical industry in recent years [16]. The remarkable anti-HIV-1 potency of BA derivatives, such as bevirimat and BMS-955176, is one of their most important properties [17,18]. The interesting anti-HIV-1 properties of BA derivatives led to the examination of their anti-influenza activity. Hong et al. [19]. reported that BA shows weak anti-influenza activity against A/PR/8/34 virus (10 μM: ~30%). Antiviral-guided isolation of the leafstalk extract of *Schefflera heptaphylla* led to the identification of two 3-*epi*-betulinic acid derivatives with anti-influenza A (H1N1) virus activity [20]. Betulinic aldehyde isolated from *Alnus japonica**,* which is used in folk remedies for influenza, exhibits certain anti-influenza effects against avian influenza KBNP-0028 (H9N2) virus with an EC_50_ value of 28.4 μM [21]. Simple modifications of BA at position C-3 or C-28 provide compounds with significant activities against influenza A virus [22,23]. However, the very poor water solubility of these compounds hampers their further development in vivo and inspires more research on better hydrophilic derivatives with potential pharmaceutical applications.

Based on these literature results of the antiviral activities of lupane-type triterpenoids and our interest in the development of natural products as potential anti-influenza agents [13,24,25], it was valuable to design and synthesize a variety of multivalent triterpenoid conjugates to disclose the relationship between their structure and activity. In our recent study, we found that one multivalent BA-α-cyclodextrin (CD) conjugate, CYY1-11, showed good anti-influenza activity (IC_50_ = 5.20 μM) against A/WSN/33 virus [25]. In the present work, we further describe the synthesis of a range of well-defined hexa-, hepta- and octavalent BA derivatives based on α-, β- and γ-CD scaffolds, respectively, with varying OEG chains (0, 1, 2, 4, 6 and 8 OEG units) as linkers via click chemistry. A total of 36 BA-CD conjugates, including compound CYY1-11 (named **51** in this manuscript), were examined to determine their anti-influenza activity against A/WSN/33 virus, and four conjugates, **51** and **69**–**71**, showed significant antiviral activities. The structure-activity relationships (SAR) of multivalent BA-CD conjugates were discussed to explore potential therapeutic agents for influenza infection.

## 2. Results and Discussion

### 2.1. Chemistry

The multivalent presentation of bioactive molecules to polymers, such as poly(ethylene glycol) (PEG), has aroused extensive interest and been widely applied in many different fields [26,27], especially in drug delivery systems [28]. Difficulties in loading a quantitative amount of drugs at a specific position of polymeric carriers, such as polymethyl methacrylate, make drug delivery systems hard to work with. To pursue our research interests in natural products with significant anti-influenza activity, we planned to synthesize a series of multivalent BA derivatives based on CD scaffolds linked by a variable length OEG chain via click chemistry. Three natural CD scaffolds and six OEG linkers were selected because of their beneficial effects on the grafted ligands, such as water solubility and good biocompatibility and immune compatibility [29].

### 2.2. Synthesis of BA-Based Alkynes ***33***–***38***

As described above, PEGs and OEGs are ubiquitously used in the pharmaceutical industry and biomedical research for the modification of proteins, peptides or nonpeptides. Therefore, OEGs were selected as the linkers between the BA pharmacophore and CD scaffold. Designed bifunctional amino alkyne linkers of different lengths (1, 2, 4, 6 and 8 OEG units) **26**–**30** were prepared from commercially available 2-azidoethanol **20**, di(ethylene glycol) **8**, tetra(ethylene glycol) **9**, hexa(ethylene glycol) **10** or octa(ethylene glycol) **11** in 27–64% yields over four steps according to conventional methods (Figure 1) [30,31]. Compound **31** could be accessed by synthesis from the cheaper precursor betulin, as reported previously [25,32]. Subsequent activation of the carboxylic acid by using TBTU/DIPEA in THF gave compound **32** in 83% yield, which was then subjected to aminolysis with commercially available propargylamine or bifunctional aminoalkyne linkers **26**–**30** to afford BA derivatives **33**–**38** bearing a terminal alkynyl group at the C-28 position (Figure 2), and their structures were characterized with ^1^H and ^13^C NMR (Appendix A). ^1^H NMR spectra of **33**–**38** showed one proton of the terminal alkynyl group at *δ* 2.41~2.43 ppm, while ^13^C NMR spectra displayed two carbons of the terminal alkynyl group at *δ* 71.06 and 80.14 ppm for **33** [33] and *δ* 74.49~74.65 and 79.44~79.58 ppm for **34**–**38**.

### 2.3. Synthesis of Multivalent BA-CD Conjugates ***69***–***86***

Multiazide-substituted CD scaffolds **48**–**50** were synthesized in 58–71% yields over three steps according to the methods described elsewhere (Figure 3) [34,35]. The copper-catalyzed 1,3-dipolar cycloaddition between each terminal alkyne-modified BA, **33**–**38**, and each multiazide-appended CD scaffold, **48**–**50**, was performed at 100 °C in the presence of sodium ascorbate and a copper sulfate catalytic system in THF/H_2_O (1:1, *v/v*) under microwave irradiation to yield a series of acetyl-protected BA-CD conjugates, **51**–**68**, in 42–55% yields, followed by a deacetylation reaction under Zemblén transesterification conditions to afford the desired homomultivalent conjugates **69**–**86** in good to excellent yields.

The structures of synthesized multivalent BA-CD conjugates **51**–**86** were characterized by NMR spectroscopy and MALDI-TOF mass spectrometry (Appendix A). Except for the signals of the linker, the ^1^H and ^13^C NMR spectra of **51**–**68** are similar to each other; therefore, only the assignment of conjugate **64** is discussed in detail as an example. The 2D ^1^H-^13^C HSQC spectrum of conjugate **64** is shown in Figure 2, and inspection of it led to the assignment of most of the peaks. In the low-field region, the signal at *δ*_H_ 7.75 ppm (Appendix A), according to the ^1^H-^13^C correlation spectrum, was assigned to triazolyl-CH. The proton of CONH at *δ*_H_ 6.13 ppm was easily identified, as there was no correlation in the 2D NMR spectrum. As a *C*_7_-symmetric macromolecular triazole adduct, conjugate **64** showed only one set of characteristic anomeric resonances [*δ*_H_ 5.49 ppm (β-CD-H_1_)]. Likewise, the other protons H_2__-__6_ of the β-CD scaffold were also assigned based on the HSQC spectrum. Two sets of peaks were clearly observed at 3.62–3.60 and 3.53–3.46 ppm (overlap with β-CD-H_4_ and NHCH_2_), which were assigned to the eleven OCH_2_ protons of the OEG group. Based on the literature data [36], major ^1^H NMR chemical shifts of the BA residue were attributed. For example, the occurrence of a vinyl residue was shown to be identified by very distinct signals at *δ*_H_ 4.72 and 4.57 ppm. Additionally, six methyl singlets were displayed at *δ*_H_ 1.67, 0.95, 0.94, 0.92, 0.80 and 0.74 ppm in the high-field region. The ^13^C NMR spectrum showed the expected number of signals (Appendix A), which were assigned with the assistance of the HSQC spectrum. In the MALDI-TOF mass spectra (Appendix A), a conjugate **64** molecular ion peak was observed at *m/z* 7228.56 (Calcd for C_385_H_616_N_28_NaO_98_^+^, 7228.25, Δ = 42.8 ppm), further confirming its identity as a fully substituted heptavalent conjugate.

The ^1^H and ^13^C NMR signals of conjugates **69**–**86** were assigned based on precursors **51**–**65**. As expected, in the majority of cases, the de-*O*-acetylation of the CD scaffold caused an upfield shift in CD-H_2_ and CD-H_3_ peaks of ~1.3 and ~1.5 ppm, respectively, but a downfield shift in both CD-C_2_ and CD-C_3_ peaks of ~2.7–3.3 ppm. For example, for conjugate **70****,** β-CD-H_2_ and β-CD-H_3_ were observed as a triplet and broad doublet at *δ* 3.45 and 3.86 ppm, respectively, which is upfield compared to the corresponding signals in conjugate **52** (*δ* 4.77 and 5.32 ppm, respectively).

### 2.4. Cytotoxicity of Multivalent BA-CD Conjugates to MDCK Cells

Cell viability assays are widely used to assess potential compound-induced toxicity. Measurement of intracellular ATP levels using ATP/luminescence readouts, such as the CellTiter-Glo reagent, is one of the most conventional and commonly used methods. Here, we evaluated the cytotoxicity of BA-CD conjugates **51**–**86** in MDCK cells before determining the anti-influenza virus activity. Culture medium containing 1% DMSO was used as a vehicle control. No significant effects on cell viability were observed with most of the multivalent BA-CD conjugates at a concentration of 100 µM, except for three conjugates **58**, **80** and **82**, which were slightly cytotoxic, with a MDCK cell viability of less than 70% (68%, 64% and 67%, respectively) (Appendix A). However, parental compound **31** possessed strong cytotoxicity towards host MDCK cells with a viability of 8.5% at the same concentration, which may due to its better cell permeability [37], encompassing the role of membrane damage in BA induced apoptosis.

### 2.5. Anti-influenza A/WSN/33 Virus Activity of Multivalent BA-CD Conjugates

Next, we employed the cytopathic effect (CPE) reduction assay to investigate the anti-influenza activity of the multivalent BA derivatives. Except for three conjugates **58**, **80** and **82** with weak cytotoxicity, the other 33 conjugates were evaluated. SAR analysis suggested that the α-CD scaffold-based conjugates exhibited higher antiviral activity against A/WSN/33 virus than the other two CD scaffold-based conjugates (e.g., **51** vs. **52** and **53**, **57** vs. **59**, and **75** vs. **76** and **77**) (Figure 3). One of the most likely reasons was that steric hindrance caused by the multiple crowded BAs inhibited the interaction between the ligand and the target protein. An exception was conjugate **81**, for which approximately 1.5-fold decreases in activity was observed compared with that of conjugate **83**. In general, the linker between BA and the CD scaffold had no obvious effect on antiviral activity. Seven conjugates **51**, **57**, **69**–**71**, **75** and **78** exhibited an inhibition rate against influenza virus A/WSN/33 (H1N1) of over 50% at a concentration of 20 μM. Further viral yield reduction studies with A/WSN/33 virus showed that they displayed dose-dependent inhibition of influenza virus replication (Table 1). Among them, conjugates **57**, **75** and **78** only showed weak anti-influenza activity with IC_50_ over 10 μM, therefore the 50% cytotoxic concentration, CC_50_, values in MDCK cells were not further determined. The other four conjugates **51** and **69**–**71** showed potent antiviral activities with IC_50_ values falling within the low micromolar range (IC_50_: 5.20–9.82 μM). More specifically, hexavalent conjugate **51** (IC_50_ of 5.20 µM) showed the highest activity and was at least 20-42 times more active than its parent compound BA (50 µg/mL: 27.6% [38], EC_50_ > 219.0 µM [39]). In addition, the CC_50_ was not determined for **51** because the dose-response curve was not achieved at the highest concentration tested (200 μM), displaying over 38.4-fold selectivity. Detailed studies on the biological activities of **51** have been described by Chen et al. [25]. The CC_50_ values of conjugates **69**–**71** in MDCK cell were also not determined because they had a value over 100 µM, in other words they were also not cytotoxic. These results indicated that the grafting of multiple BAs onto the primary face of CD scaffolds was an effective strategy for enhancing the anti-influenza activity of BAs.

During the last two decades, BA and its derivatives have attracted special interest due to their remarkable anti-HIV-1 activity with three derivatives (bevirimat [40], BMS-955176 [41] and GSK-2838232 [42]) entering clinical trials. In recent years, an increasing number of studies with regard to their potential applications against other viruses have been performed [43,44,45]. As a class of anti-HIV-1 agents with new mechanisms of action (entry [17] and maturation [41]), however, the anti-influenza mechanism of action of BA and its derivatives has not yet been clearly elucidated. The primary study of the antiviral mechanism of **51** based on surface plasmon resonance assay indicated that multivalent BA derivatives can bind specifically with influenza HA protein with K_D_ value of 1.50 μM [25], thus blocking influenza virus entry into host cells. Compared to **51**, some conjugates, such as **81** and **83**, showed relative weak binding affinity to influenza HA protein with K_D_ values over 10 μM (Appendix A), which agreed well with their anti-influenza activities. Further efforts to uncover the function of the HA protein binding domain of multivalent BA-CD conjugates and to investigate how the structural features contribute to the binding domain will provide important insights into the multivalent binding mechanism and help to guide the design of more effective multivalent pentacyclic triterpene derivatives targeting this important protein.

## 3. Materials and Methods

### 3.1. Materials

α-, β- and γ-CD were purchased from Kaiguo Science & Technology Co., Ltd. (Beijing, China). BA was purchased from Bide Pharmatech. Ltd (Beijing, China). All the other chemical reagents and solvents were commercially available and used as received. The MDCK cell line was obtained from Crown Bioscience Inc. (San Diego, CA, USA), which were grown in Dulbecco’s Modified Eagle Medium (DMEM; Gibco BRL Life Technologies Inc., Grand Island, NY, USA) and supplemented with 10% fetal bovine serum (FBS; PAA Laboratories, Pasching, Austria) at 37 °C in a humidified atmosphere of 5% CO_2_. The CellTiter-Glo luminescent cell viability assay kit was purchased from Promega Corp. (Madison, WI, USA). 

The NMR spectra were obtained on Bruker 400 and 600 MHz spectrometers (Bruker Daltonics., Billerica, MA, USA). The value of chemical shifts (*δ*) are given in ppm and coupling constants (*J*) in hertz (Hz). High-resolution electrospray mass spectra (HRMS) and MALDI-TOF-MS were recorded by a Bruker APEX IV FT_MS (7.0 T) mass spectrometer (Bruker Daltonics Inc., Billerica, MA, USA) and an AB Sciex TOF/TOF™ 72115 mass spectrometer (AB Sciex, Redwood City, CA, USA), respectively. The reaction progress and chromatography fractions were monitored by analytical thin-layer chromatography (TLC) on 0.25 mm thickness E. Merck pre-coated plates of silica gel 60 F_254_. The spots were visualized by immersion of the TLC plate in an appropriate solution followed by heating with a hot gun. The following staining solutions were applied: ninhydrin staining solution [ninhydrin (10.0 g) and ethanol (300 mL)], cerium molybdate staining solution [Ce(NH_4_)_2_(NO_3_)_6_ (0.5 g, 0.9 mmol), (NH_4_)_6_Mo_7_O_24_·4H_2_O (24.0 g, 19.4 mmol), concentrated aqueous H_2_SO_4_ (30 mL) and H_2_O (470 mL)]. Column chromatography was performed on silica gel (200–300 mesh). Linkers **26**–**30** [30,31], BA derivatives **32**–**33 [33]** and hexavalent BA-α-CD conjugate **51** [25] were synthesized by literature methods, and the data were consistent with those published.

### 3.2. General Procedure A for the Synthesis of Terminal Propargylated OEG-Tethered BA Derivatives (***34***–***38***)

Na_2_CO_3_ (0.68 mmol, 2.0 equiv.) was added to a solution of 1-benzotriazolyl 3β-hydroxy-lup-20(29)-en-28-oate (**32**) (0.34 mmol, 1.0 equiv.) and terminal propargylated OEG-tethered amine (**26**–**30**) (0.41 mmol, 1.2 equiv.) in DMF (4 mL), and the mixture was stirred at room temperature for 24 h. After completion of the reaction, as indicated by TLC, the solvent was evaporated under reduced pressure, and the obtained residue was extracted with CH_2_Cl_2_ (10 mL × 3), dried with anhydrous Na_2_SO_4_, filtered, and evaporated. The pure product was obtained by column chromatography performed on silica gel.

#### 3.2.1. Synthesis of *N*-(2-(2-Propyn-1-yloxy)ethyl)-3β-hydroxy-lup-20(29)-en-28-amide (**34**)

Prepared from **32** and 2-(propyn-1-yloxy)-ethanamine (**26**) according to general procedure A, the residue was purified by flash chromatography (eluent: CH_2_Cl_2_:(CH_3_)_2_CO = 40:1) to afford **34** as a white product with a yield of 93%. R_f_ = 0.36 (CH_2_Cl_2_:(CH_3_)_2_CO = 20:1); ^1^H NMR (400 MHz, CDCl_3_): *δ* 5.96 (t, 1H, *J* = 5.6 Hz), 4.73 (d, 1H, *J* = 1.7 Hz), 4.58 (s, 1H), 4.15 (d, 2H, *J* = 2.4 Hz), 3.62–3.38 (m, 4H), 3.17 (dd, 1H, *J* = 10.5, 5.6 Hz), 3.11 (dt, 1H, *J* = 11.5, 4.6 Hz), 2.46–2.39 (m, 2H), 2.00–0.85 (m, other aliphatic ring protons), 1.68, 0.96, 0.95, 0.93, 0.81, 0.75 (s, each 3H, 6 × CH_3_), 0.67 (d, 1H, *J* = 9.0 Hz); ^13^C NMR (100 MHz, CDCl_3_): *δ* 176.19, 150.95, 109.32, 79.45, 78.96, 74.64, 69.07, 58.25, 55.71, 55.37, 50.61, 50.07, 46.82, 42.48, 40.74, 38.84, 38.79, 38.71, 38.34, 37.78, 37.19, 34.39, 33.68, 30.88, 29.42, 27.97, 27.41, 25.61, 20.91, 19.46, 18.29, 16.12, 15.34, 14.64; ESI-HRMS (*m/z*) Calcd for C_35_H_56_NO_3_ [M + H]^+^: 538.4255. Found 538.4247.

#### 3.2.2. Synthesis of *N*-(2-(2-(2-Propyn-1-yloxy)ethoxy)ethyl)-3β-hydroxy-lup-20(29)-en-28-amide (**35**)

Prepared from **32** and 2-(2-(2-propyn-1-yloxy)ethoxy)-ethanamine (**27**) according to general procedure A, the residue was purified by flash chromatography (eluent: PE:EtOAc = 2:1) to afford **35** as a white product with a yield of 58%. R_f_ = 0.16 (PE:EtOAc = 2:1); ^1^H NMR (400 MHz, CDCl_3_): *δ* 6.05 (t, 1H, *J* = 5.6 Hz), 4.72 (s, 1H), 4.58 (s, 1H), 4.19 (d, 2H, *J* = 2.4 Hz), 3.69–3.62 (m, 4H), 3.56–3.36 (m, 4H), 3.17 (dd, 1H, *J* = 11.4, 5.0 Hz), 3.11 (dt, 1H, *J* = 12.7, 5.8 Hz), 2.45–2.38 (m, 2H), 1.98–0.85 (m, other aliphatic ring protons), 1.67, 0.96, 0.95, 0.92, 0.80, 0.74 (s, each 3H, 6 × CH_3_), 0.66 (d, 1H, *J* = 9.0 Hz); ^13^C NMR (100 MHz, CDCl_3_): *δ* 176.18, 150.98, 109.28, 79.44, 78.94, 74.65, 70.08, 69.98, 68.96, 58.37, 55.69, 55.37, 50.60, 50.08, 46.81, 42.46, 40.73, 38.84, 38.78, 38.70, 38.33, 37.76, 37.19, 34.39, 33.65, 30.89, 29.40, 27.97, 27.41, 25.61, 20.91, 19.46, 18.29, 16.14, 16.11, 15.34, 14.63; ESI-HRMS (*m/z*) Calcd for C_37_H_60_NO_4_ [M + H]^+^: 582.4517. Found 582.4510.

#### 3.2.3. Synthesis of *N*-(3,6,9,12-Tetraoxapentadec-14-yn-1-yl)-3β-hydroxy-lup-20(29)-en-28-amide (**36**)

Prepared from **32** and 3,6,9,12-tetraoxapentadec-14-yn-1-amine (**28**) according to general procedure A, the residue was purified by flash chromatography (eluent: PE:EtOAc = 1:3) to afford **36** as a white product with a yield of 69%. R_f_ = 0.30 (PE:EtOAc = 1:3); ^1^H NMR (400 MHz, CDCl_3_): *δ* 6.06 (s, 1H), 4.72 (s, 1H), 4.57 (s, 1H), 4.19 (s, 2H), 3.68–3.40 (m, 16H), 3.13 (m, 2H), 2.42 (t, 2H, *J* = 13.0 Hz), 1.94 (m, 2H), 1.77–0.95 (m, other aliphatic ring protons), 1.67 (s, 3H), 0.95 (s, 6H, 2 × CH_3_ ), 0.92 (s, 3H), 0.86 (d, 1H, *J* = 12.6 Hz), 0.86, 0.74 (s, each 3H, 2 × CH_3_ ), 0.66 (d, 1H, *J* = 7.7 Hz); ^13^C NMR (100 MHz, CDCl_3_): *δ* 176.15, 150.97, 109.26, 79.58, 78.91, 74.51, 70.59, 70.51, 70.40, 70.20, 70.05, 69.07, 58.37, 55.65, 55.36, 50.60, 50.09, 46.78, 42.44, 40.72, 38.82, 38.69, 38.31, 37.72, 37.17, 34.39, 33.62, 30.88, 29.39, 27.96, 27.40, 25.60, 20.90, 19.46, 18.28, 16.14, 16.10, 15.33, 14.62; ESI-HRMS (*m/z*) Calcd for C_41_H_71_N_2_O_6_ [M + NH_4_]^+^: 687.5307. Found 687.5324.

#### 3.2.4. Synthesis of *N*-(3,6,9,12,15,18-Hexaoxaheneicos-20-yn-1-yl)-3β-hydroxy-lup-20(29)-en-28-amide (**37**)

Prepared from **32** and 3,6,9,12,15,18-hexaoxaheneicos-20-yn-1-amine (**29**) according to general procedure A, the residue was purified by flash chromatography (eluent: PE:EtOAc = 1:3) to afford **37** as a colourless oil with a yield of 61%. R_f_ = 0.17 (PE:EtOAc = 1:3); ^1^H NMR (400 MHz, CDCl_3_): *δ* 6.12 (t, 1H, *J* = 5.5 Hz), 4.71 (d,1H, *J* = 1.9 Hz), 4.56 (s, 1H), 4.18 (d, 2H, *J* = 2.4 Hz), 3.69–3.57 (m, 20H), 3.51 (t, 2H, *J* = 4.8 Hz), 3.48–3.42 (m, 1H), 3.40–3.33 (m, 1H), 3.15 (dd, 1H, *J* = 11.2, 5.1 Hz), 3.10 (dt, 1H, *J* = 11.6, 4.7 Hz), 2.45–2.38 (m, 2H), 1.97–1.87 (m, 2H), 1.77–0.79 (m, other aliphatic ring protons), 1.66 (s, 3H, CH_3_), 0.94 (s, 6H, 2 × CH_3_), 0.91, 0.79, 0.73 (each 3H, 3 × CH_3_), 0.65 (d, 1H, *J* = 8.9 Hz); ^13^C NMR (100 MHz, CDCl_3_): *δ* 176.14, 150.95, 109.22, 79.59, 78.85, 74.49, 70.51, 70.43, 70.34, 70.14, 70.05, 69.03, 58.33, 55.60, 55.32, 50.56, 50.06, 46.73, 42.39, 40.68, 38.79, 38.66, 38.28, 37.67, 37.14, 34.36, 33.55, 30.85, 29.35, 27.93, 27.36, 25.57, 20.87, 19.43, 18.25, 16.11, 16.07, 15.31, 14.59; ESI-HRMS (*m/z*) Calcd for C_45_H_79_N_2_O_8_ [M + NH_4_]^+^: 775.5831. Found 775.5829.

#### 3.2.5. Synthesis of *N*-(3,6,9,12,15,18,21,24-Octaoxaheneicos-26-Yn-1-Yl)-3β-Hydroxy-lup-20(29)-en-28-amide (**38**)

Prepared from **32** and 3,6,9,12,15,18,21,24-octaoxaheptacos-26-yn-1-amine (**30**) according to general procedure A, the residue was purified by flash chromatography (eluent: PE:EtOAc = 1:3) to afford **38** as a colourless oil with a yield of 64%. R_f_ = 0.16 (PE:EtOAc = 1:3); ^1^H NMR (400 MHz, CDCl_3_): *δ* 6.08 (s, 1H), 4.69 (s, 1H), 4.54 (s, 1H), 4.17 (d, 2H, *J* = 1.5 Hz), 3.66–3.58 (m, 28H), 3.50–3.33 (m, 4H), 3.14 (dd, 1H, *J* = 10.7, 4.8 Hz), 3.12–3.06 (m, 1H), 2.42–2.32 (m, 2H), 1.94–1.86 (m, 2H), 1.75–1.70 (m, 1H), 1.64–0.82 (m, other aliphatic ring protons), 1.64 (s, 3H, 1 × CH_3_), 0.92 (s, 6H, 2 × CH_3_), 0.89, 0.77, 0.71 (s, each 3H, 3 × CH_3_), 0.63 (d, 1H, *J* = 8.7 Hz); ^13^C NMR (100 MHz, CDCl_3_): 176.07, 150.90, 109.20, 79.52, 78.75, 74.51, 70.44, 70.37, 70.28, 70.08, 69.96, 68.96, 58.28, 55.53, 55.23, 50.46, 49.95, 46.67, 42.32, 40.59, 38.73, 38.58, 38.23, 37.60, 37.05, 34.26, 33.50, 30.76, 29.28, 27.89, 27.28, 25.47, 20.79, 19.37, 18.18, 16.04, 15.29, 14.54; ESI-HRMS (*m/z*) Calcd for C_49_H_87_N_2_O_10_ [M + NH_4_^+^]^+^: 863.6361. Found 863.6345.

### 3.3. General procedure B for the Synthesis of Multivalent BA-CD Conjugates (***52***–***68***)

CuSO_4_·5H_2_O (0.10 mmol, 1.0 equiv.) and sodium *L*-ascorbate (1.1 equiv. per mol azide) were added to a solution of multiazide-substituted α-, *β*- and γ-CD (**48**–**50**) (0.10 mmol, 1.0 equiv.) and terminal propargylated OEG-tethered BA derivatives (**33**–**38**) (1.1 equiv. per mol azide) in THF-H_2_O (10 mL, *v/v* 1:1). The reaction vessel was placed in a vigorously stirred CEM Discover SP microwave reactor (100 °C, 50 W) and heated for 1 h. After the reaction mixture was extracted with CH_2_Cl_2_ (10 mL × 3), it was washed with water and brine and dried with anhydrous Na_2_SO_4_. The solvent was evaporated under reduced pressure, and the crude product was purified by column chromatography.

#### 3.3.1. Synthesis of Heptakis 6-Deoxy-6-[4-*N*-(3β-Hydroxy-lup-20(29)-en-28-oyl)-aminomethyl-1*H*-1,2,3-triazol-1-yl]-2,3-di-*O*-acetyl-β-CD (**52**)

Prepared from **33** and **49** according to general procedure B, the residue was purified by flash chromatography (eluent: CH_2_Cl_2_:CH_3_OH = 12:1) to afford **52** as a white foam with a yield of 28%. R_f_ = 0.40 (CH_2_Cl_2_:CH_3_OH = 8:1); ^1^H NMR (400 MHz, CDCl_3_): *δ* 7.66 (s, 7H), 6.75 (s, 7H), 5.44 (s, 7H), 5.37–5.27 (m, 7H), 4.94 (d, 7H, *J* = 14.1 Hz), 4.81–4.54 (m, 35H), 4.74 (s, 7H), 4.19–4.01 (m, 7H), 3.53 (t, 7H, *J* = 8.0 Hz), 3.21–3.05 (m, 14H), 2.48 (m, 7H), 2.07 (s, 21H), 2.02 (s, 21H), 1.91–0.71 (m, other aliphatic ring protons), 1.65, 0.95, 0.95, 0.92, 0.81, 0.76 (s, each 21H, 42 × CH_3_), 0.67 (d, 7H, *J* = 8.4 Hz); ^13^C NMR (150 MHz, CDCl_3_): *δ* 176.47, 170.33, 169.37, 150.75, 145.45, 124.48, 109.53, 96.57, 78.91, 70.42, 69.96, 69.65, 55.57, 55.33, 50.61, 50.07, 46.60, 42.45, 40.73, 38.83, 38.70, 38.18, 37.59, 37.18, 34.82, 34.43, 33.52, 30.79, 27.96, 27.38, 27.18, 25.58, 20.91, 20.72, 20.65, 19.37, 18.30, 16.14, 15.37, 14.63; MALDI-TOF MS (*m**/z*) Calcd for C_301_H_448_N_28_NaO_56_ [M + Na]^+^: 5374.30. Found 5374.64.

#### 3.3.2. Synthesis of Octakis 6-Deoxy-6-[4-*N*-(3β-Hydroxy-lup-20(29)-en-28-oyl)-aminomethyl-1*H*-1,2,3-triazol-1-yl]-2,3-di-*O*-acetyl-γ-CD (**53**)

Prepared from **33** and **50** according to general procedure B, the residue was purified by flash chromatography (eluent: CH_2_Cl_2_:CH_3_OH = 10:1) to afford **53** as a white foam with a yield of 31%. R_f_ = 0.45 (CH_2_Cl_2_:CH_3_OH = 8:1); ^1^H NMR (400 MHz, CDCl_3_): *δ* 7.64 (s, 8H), 6.69 (s, 8H), 5.45 (s, 8H), 5.33 (t, 8H, *J* = 8.7 Hz), 4.85 (d, 8H, *J* = 13.6 Hz), 4.87–4.56 (m, 48H), 4.42 (s, 8H, *J* = 7.2 Hz), 4.08 (d, 8H, *J* = 14.9 Hz), 3.51 (t, 8H, *J* = 8.6 Hz), 3.17 (dd, 8H, *J* = 10.4, 3.2 Hz), 3.09 (dt, 8H, *J* = 11.0, 4.2 Hz), 2.47 (t, 8H, *J* = 10.0 Hz), 2.06 (s, 24H), 2.05 (m, 8H), 2.03 (s, 24H), 1.87–0.89 (m, other aliphatic ring protons), 1.64 (s, 24H, 8 × CH_3_), 0.95 (s, 48H, 16 × CH_3_), 0.92, 0.80, 0.74 (s, each 24H, 24 × CH_3_), 0.67 (d, 8H, *J* = 9.7 Hz); ^13^C NMR (150 MHz, CDCl_3_): *δ* 176.48, 170.29, 169.44, 150.76, 145.39, 124.56, 109.56, 96.22, 78.92, 76.04, 70.27, 69.95, 55.61, 55.34, 50.61, 50.09, 49.91, 46.65, 42.46, 40.75, 38.84, 38.71, 38.21, 37.63, 37.19, 34.85, 34.44, 33.51, 30.83, 29.52, 27.39, 25.58, 20.93, 20.78, 20.64, 19.36, 18.31, 16.16, 15.39, 14.64; MALDI-TOF MS (*m/z*) Calcd for C_344_H_512_N_32_NaO_64_ [M + Na]^+^: 6143.03. Found 6143.16.

#### 3.3.3. Synthesis of Hexakis 6-Deoxy-6-[4-*N*-[(2-(2-Propyn-1-yloxy)ethyl)-3β-hydroxy-lup-20(29)-en-28-oyl]-aminomethyl-1*H*-1,2,3-triazol-1-yl]-2,3-di-*O*-acetyl-α-CD (**54**)

Prepared from **34** and **48** according to general procedure B, the residue was purified by flash chromatography (eluent: CH_2_Cl_2_:CH_3_OH = 20:1) to afford **54** as a white foam with a yield of 61%. R_f_ = 0.38 (CH_2_Cl_2_:CH_3_OH = 10:1); ^1^H NMR (600 MHz, CDCl_3_): *δ* 7.64 (s, 6H), 6.50 (s, 6H), 5.51 (s, 1H), 5.40 (s, 6H), 4.71–4.52 (m, 48H), 3.56–3.38 (m, 30H), 3.17 (m, 6H), 3.10 (m, 6H), 2.49 (m, 6H), 2.04–2.00 (m, 42H), 1.89–0.86 (m, other aliphatic ring protons), 1.66 (s, 18H, 6 × CH_3_), 0.95 (s, 36H, 12 × CH_3_), 0.93, 0.80, 0.74 (s, each 18H, 18 × CH_3_), 0.66 (d, 6H, *J* = 8.6 Hz); ^13^C NMR (150 MHz, CDCl_3_): *δ* 176.55, 170.32, 169.07, 150.93, 144.56, 125.55, 109.36, 96.40, 78.90, 70.63, 70.02, 69.69, 64.17, 55.58, 55.36, 50.61, 50.15, 46.67, 42.41, 40.73, 39.05, 38.83, 38.69, 38.29, 37.57, 37.18, 34.41, 33.56, 30.88, 29.41, 27.99, 27.39, 25.59, 20.94, 20.69, 19.47, 18.31, 16.19, 16.14, 15.37, 14.63; MALDI-TOF MS (*m/z*) Calcd for C_270_H_408_N_24_NaO_54_ [M + Na]^+^: 4877.34. Found 4877.08.

#### 3.3.4. Synthesis of Heptakis 6-Deoxy-6-[4-*N*-[(2-(2-Propyn-1-yloxy)ethyl)-3β-hydroxy-lup-20(29)-en-28-oyl]-aminomethyl-1*H*-1,2,3-triazol-1-yl]-2,3-di-*O*-acetyl-β-CD (**55**)

Prepared from **34** and **49** according to general procedure B, the residue was purified by flash chromatography (eluent: CH_2_Cl_2_:CH_3_OH = 15:1) to afford **55** as a white foam with a yield of 20%. R_f_ = 0.29 (CH_2_Cl_2_:CH_3_OH = 10:1); ^1^H NMR (600 MHz, CDCl_3_): *δ* 7.71 (s, 7H), 6.46 (s, 7H), 5.46 (s, 7H), 5.34 (m, 7H), 4.89–4.44 (m, 56H), 3.54–3.36 (m, 35H), 3.16 (dd, 7H, *J* = 11.3, 4.4 Hz), 3.10 (dt, 7H, *J* = 10.9, 4.2 Hz), 2.49 (m, 7H), 2.06–2.00 (m, 49H), 1.90–0.86 (m, other aliphatic ring protons), 1.66 (s, 21H, 7 × CH_3_), 0.94 (s, 42H, 14 × CH_3_), 0.92, 0.80, 0.74 (s, each 21H, 21 × CH_3_), 0.66 (d, 7H, *J* = 8.9 Hz); ^13^C NMR (150 MHz, CDCl_3_): *δ* 176.52, 170.33, 169.30, 150.92, 144.80, 125.45, 109.38, 96.45, 78.89, 70.34, 69.83, 69.67, 64.15, 55.60, 55.35, 50.60, 50.13, 46.68, 42.41, 40.73, 39.00, 38.83, 38.69, 38.32, 37.57, 37.18, 34.41, 33.56, 30.86, 29.42, 27.99, 27.39, 25.58, 22.65, 20.93, 20.72, 20.67, 19.45, 18.31, 16.17, 16.14, 15.37, 14.63; MALDI-TOF MS (*m/z*) Calcd for C_315_H_476_N_28_NaO_63_ [M + Na]^+^: 5686.40. Found 5686.94.

#### 3.3.5. Synthesis of Octakis 6-Deoxy-6-[4-*N*-[(2-(2-Propyn-1-yloxy)ethyl)-3β-hydroxy-lup-20(29)-en-28-oyl]-aminomethyl-1*H*-1,2,3-triazol-1-Yl]-2,3-di-*O*-acetyl-γ-CD (**56**)

Prepared from **34** and **50** according to general procedure B, the residue was purified by flash chromatography (eluent: CH_2_Cl_2_:CH_3_OH = 15:1) to afford **56** as a white foam with a yield of 34%. R_f_ = 0.15 (CH_2_Cl_2_:CH_3_OH = 10:1); ^1^H NMR (600 MHz, CDCl_3_): *δ* 7.75 (s, 8H), 6.48 (s, 8H), 5.48 (s, 8H), 5.35 (m, 8H), 4.83–4.40 (m, 64H), 3.56–3.39 (m, 40H), 3.17 (d, 8H, *J* = 7.0 Hz), 3.10 (dt, 8H, *J* = 10.9, 4.3 Hz), 2.49 (t, 8H, *J* = 9.7 Hz), 2.07–2.02 (m, 56H), 1.92–0.86 (m, other aliphatic ring protons), 1.66 (s, 24H, 8 × CH_3_), 0.95 (s, 48H, 16 × CH_3_), 0.93, 0.80, 0.74 (s, each 24H, 24 × CH_3_), 0.66 (d, 8H, *J* = 8.9 Hz); ^13^C NMR (150 MHz, CDCl_3_): *δ* 176.54, 170.26, 169.40, 150.93, 144.85, 125.54, 109.38, 96.18, 78.90, 69.97, 69.65, 64.24, 55.61, 55.36, 50.61, 50.13, 49.91, 46.68, 42.42, 40.74, 38.93, 38.84, 38.70, 38.34, 37.58, 37.19, 34.41, 33.57, 30.88, 29.42, 28.00, 27.40, 25.60, 20.95, 20.75, 20.65, 19.46, 18.32, 16.18, 16.15, 15.39, 14.64; MALDI-TOF MS (*m/z*) Calcd for C_360_H_545_N_32_O_72_ [M + H]^+^: 6473.47. Found 6473.59.

#### 3.3.6. Synthesis of Hexakis 6-Deoxy-6-[4-*N*-[(2-(2-(2-Propyn-1-yloxy)ethoxy)ethyl)-3β-hydroxy-lup-20(29)-en-28-oyl]-aminomethyl-1*H*-1,2,3-triazol-1-yl]-2,3-di-*O*-acetyl-α-CD (**57**)

Prepared from **35** and **48** according to general procedure B, the residue was purified by flash chromatography (eluent: CH_2_Cl_2_:CH_3_OH = 20:1) to afford **57** as a white foam with a yield of 74%. R_f_ = 0.40 (CH_2_Cl_2_:CH_3_OH = 10:1); ^1^H NMR (600 MHz, CDCl_3_): *δ* 7.73 (s, 6H), 6.24 (t, 6H, *J* = 5.5 Hz), 5.49 (t, 6H, *J* = 9.5 Hz), 5.41 (d, 6H, *J* = 3.1 Hz), 4.71–4.50 (m, 48H), 3.64–3.34 (m, 54H), 3.16 (d, 6H, *J* = 11.0 Hz), 3.10 (dt, 6H, *J* = 11.0, 4.3 Hz), 2.47–2.43 (m, 6H), 2.03–1.99 (m, 42H), 1.93–1.88 (m, 6H), 1.78–0.86 (m, other aliphatic ring protons), 1.66 (s, 18H, 6 × CH_3_), 0.95 (s, 36H, 12 × CH_3_), 0.92, 0.80, 0.74 (s, each 18H, 18 × CH_3_), 0.66 (d, 6H, *J* = 9.4 Hz); ^13^C NMR (150 MHz, CDCl_3_): *δ* 176.30, 170.30, 169.07, 150.96, 144.70, 125.63, 109.33, 96.74, 78.89, 70.83, 70.06, 70.01, 69.94, 64.40, 55.62, 55.34, 50.58, 50.09, 46.73, 42.43, 40.72, 38.83, 38.75, 38.68, 38.33, 37.65, 37.17, 34.40, 33.57, 30.87, 29.40, 27.98, 27.39, 25.59, 20.91, 20.71, 19.45, 18.30, 16.16, 16.13, 15.37, 14.63; MALDI-TOF MS (*m/z*) Calcd for C_282_H_432_N_24_NaO_60_ [M + Na]^+^: 5141.66. Found 5141.23.

#### 3.3.7. Synthesis of Heptakis 6-Deoxy-6-[4-*N*-[(2-(2-(2-Propyn-1-yloxy)ethoxy)ethyl)-3β-hydroxy-lup-20(29)-en-28-Oyl]-aminomethyl-1*H*-1,2,3-triazol-1-Yl]-2,3-di-*O*-acetyl-β-CD (**58**)

Prepared from **35** and **49** according to general procedure B, the residue was purified by flash chromatography (eluent: CH_2_Cl_2_:CH_3_OH = 15:1) to afford **58** as a white foam with a yield of 60%. R_f_ = 0.39 (CH_2_Cl_2_:CH_3_OH = 10:1); ^1^H NMR (600 MHz, CDCl_3_): *δ* 7.76 (s, 7H), 6.23 (t, 7H, *J* = 5.5 Hz), 5.49 (s, 7H), 5.35 (t, 7H, *J* = 8.6 Hz), 4.87–4.45 (m, 56H), 3.64–3.34 (m, 63H), 3.16 (dd, 7H, *J* = 11.5, 4.5 Hz), 3.10 (dt, 7H, *J* = 11.1, 4.4 Hz), 2.44 (dt, 7H, *J* = 12.7, 3.3 Hz), 2.05–1.88 (m, 56H), 1.77–0.84 (m, other aliphatic ring protons), 1.66, 0.95, 0.94, 0.92, 0.80, 0.73 (s, each 21H, 42 × CH_3_), 0.66 (d, 7H, *J* = 9.4 Hz); ^13^C NMR (150 MHz, CDCl_3_): *δ* 176.30, 170.35, 169.32, 150.93, 144.86, 125.53, 109.35, 96.33, 78.88, 76.53, 70.54, 70.06, 70.04, 69.90, 69.71, 64.39, 55.63, 55.34, 50.58, 50.09, 46.74, 42.43, 40.72, 38.83, 38.79, 38.69, 38.34, 37.66, 37.18, 34.40, 33.59, 30.88, 29.40, 27.99, 27.39, 25.59, 20.91, 20.72, 20.67, 19.46, 18.30, 16.16, 16.13, 15.38, 14.64; MALDI-TOF MS (*m/z*) Calcd for C_329_H_504_N_28_NaO_70_ [M + Na]^+^: 5994.77. Found 5994.38.

#### 3.3.8. Synthesis of Octakis 6-Deoxy-6-[4-*N*-[(2-(2-(2-Propyn-1-yloxy)ethoxy)ethyl)-3β-hydroxy-lup-20(29)-en-28-oyl]-aminomethyl-1*H*-1,2,3-triazol-1-yl]-2,3-di-*O*-acetyl-γ-CD (**59**)

Prepared from **35** and **50** according to general procedure B, the residue was purified by flash chromatography (eluent: CH_2_Cl_2_:CH_3_OH = 15:1) to afford **59** as a white foam with a yield of 68%. R_f_ = 0.33 (CH_2_Cl_2_:CH_3_OH = 10:1); ^1^H NMR (600 MHz, CDCl_3_): *δ* 7.77 (s, 8H), 6.24 (t, 8H, *J* = 5.5 Hz), 5.52 (s, 8H), 5.35 (t, 8H, *J* = 8.7 Hz), 4.79–4.44 (m, 64H), 3.65–3.34 (m, 72H), 3.16 (d, 8H, *J* = 10.9 Hz), 3.10 (dt, 8H, *J* = 11.0, 4.4 Hz), 2.44 (dt, 8H, *J* = 12.6, 3.2 Hz), 2.05–1.97 (m, 56H), 1.95–0.86 (m, other aliphatic ring protons), 1.66, 0.95, 0.94, 0.91, 0.79, 0.73 (s, each 24H, 48 × CH_3_), 0.66 (d, 8H, *J* = 9.4 Hz); ^13^C NMR (150 MHz, CDCl_3_): *δ* 176.29, 170.26, 169.42, 150.92, 144.85, 125.58, 109.33, 95.98, 78.85, 75.71, 70.25, 70.06, 70.03, 69.88, 69.82, 64.40, 55.62, 55.33, 50.57, 50.08, 46.73, 42.42, 40.71, 38.82, 38.78, 38.68, 38.33, 37.65, 37.16, 34.39, 33.57, 30.87, 29.39, 27.98, 27.38, 25.58, 20.90, 20.76, 20.63, 19.45, 18.29, 16.15, 16.12, 15.38, 14.63; MALDI-TOF MS (*m/z*) Calcd for C_376_H_577_N_32_O_80_ [M + H]^+^: 6825.90. Found 6824.69.

#### 3.3.9. Synthesis of Hexakis 6-Deoxy-6-[4-*N*-[(3,6,9,12-Tetraoxapentadec-14-yn-1-yl)-3β-hydroxy-lup-20(29)-en-28-oyl]-aminomethyl-1*H*-1,2,3-triazol-1-yl]-2,3-di-*O*-acetyl-α-CD (**60**)

Prepared from **36** and **48** according to general procedure B, the residue was purified by flash chromatography (eluent: CH_2_Cl_2_:CH_3_OH = 7:1) to afford **60** as a white foam with a yield of 39%. R_f_ = 0.31 (CH_2_Cl_2_:CH_3_OH = 10:1); ^1^H NMR (600 MHz, CDCl_3_): *δ* 7.68 (s, 6H), 6.11 (t, 6H, *J* = 5.3 Hz), 5.46 (t, 6H, *J* = 9.1 Hz), 5.40 (d, 6H, *J* = 2.8 Hz), 4.64 (m, 48H), 3.66–3.50 (m, 84H), 3.45 (m, 6H), 3.37 (m, 6H), 3.16 (dd, 6H, *J* = 11.5, 4.6 Hz), 3.10 (dt, 6H, *J* = 11.1, 4.4 Hz), 2.42 (m, 6H), 2.07–1.90 (m, 48H), 1.75 (m, 6H), 1.68–0.97 (m, other aliphatic ring protons), 1.66, 0.95, 0.94, 0.91, 0.80, 0.73 (s, each 18H, 36 × CH_3_), 0.87 (m, 6H), 0.66 (d, 6H, *J* = 9.2 Hz); ^13^C NMR (150 MHz, CDCl_3_): *δ* 176.17, 170.25, 168.99, 150.93, 144.64, 125.60, 109.29, 96.69, 78.85, 70.94, 70.47, 70.43, 70.14, 70.02, 69.94, 69.79, 64.39, 55.63, 55.34, 50.58, 50.51, 50.08, 46.77, 42.43, 40.71, 38.82, 38.68, 38.30, 37.70, 37.16, 34.38, 33.60, 30.87, 29.64, 29.38, 29.26, 27.98, 27.38, 27.15, 25.58, 20.89, 20.71, 20.67, 19.45, 18.28, 16.14, 16.11, 15.36, 14.63; MALDI-TOF MS (*m/z*) Calcd for C_306_H_481_N_24_O_72_ [M + H]^+^: 5648.31. Found 5648.34.

#### 3.3.10. Synthesis of Heptakis 6-Deoxy-6-[4-*N*-[(3,6,9,12-Tetraoxapentadec-14-yn-1-yl)-3β-hydroxy-lup-20(29)-en-28-oyl]-aminomethyl-1*H*-1,2,3-triazol-1-yl]-2,3-di-*O*-acetyl-β-CD (**61**)

Prepared from **36** and **49** according to general procedure B, the residue was purified by flash chromatography (eluent: CH_2_Cl_2_:CH_3_OH = 10:1) to afford **61** as a white foam with a yield of 31%. R_f_ = 0.28 (CH_2_Cl_2_:CH_3_OH = 10:1); ^1^H NMR (600 MHz, CDCl_3_): *δ* 7.75 (s, 7H), 6.12 (s, 7H), 5.49 (s, 7H), 5.35 (m, 7H), 4.87–4.47 (m, 56H), 3.63 (m, 84H), 3.52 (m, 28H), 3.37 (m, 7H), 3.16 (dd, 7H, *J* = 11.5, 4.6 Hz), 3.10 (dt, 7H, *J* = 11.2, 4.4 Hz), 2.42 (dt, 7H, *J* = 12.8, 3.2 Hz), 1.99 (m, 56H), 1.75 (m, 7H), 1.66–1.11 (m, other aliphatic ring protons), 1.66, 0.95, 0.94, 0.92, 0.80, 0.73 (s, each 21H, 42 × CH_3_), 0.87 (m, 7H), 0.66 (d, 7H, *J* = 9.3 Hz); ^13^C NMR (150 MHz, CDCl_3_): *δ* 176.19, 170.33, 169.33, 150.93, 144.86, 125.53, 109.31, 96.29, 78.86, 76.47, 70.48, 70.44, 70.39, 70.15, 70.04, 69.92, 69.65, 64.41, 55.64, 55.35, 50.59, 50.09, 49.96, 46.78, 42.44, 40.72, 38.83, 38.69, 38.31, 37.72, 37.17, 34.39, 33.60, 29.65, 29.39, 29.27, 27.98, 27.39, 27.16, 25.59, 25.50, 20.90, 20.71, 20.67, 19.46, 18.29, 16.15, 16.11, 15.37, 14.64; MALDI-TOF MS (*m/z*) Calcd for C_357_H_560_N_28_NaO_84_ [M + Na]^+^: 6611.51. Found 6611.99.

#### 3.3.11. Synthesis of Octakis 6-Deoxy-6-[4-*N*-[(3,6,9,12-Tetraoxapentadec-14-yn-1-yl)-3β-hydroxy-lup-20(29)-en-28-oyl]-aminomethyl-1*H*-1,2,3-triazol-1-yl]-2,3-di-*O*-acetyl-γ-CD (**62**)

Prepared from **36** and **50** according to general procedure B, the residue was purified by flash chromatography (eluent: CH_2_Cl_2_:CH_3_OH = 7:1) to afford **62** as a white foam with a yield of 56%. R_f_ = 0.38 (CH_2_Cl_2_:CH_3_OH = 10:1); ^1^H NMR (600 MHz, CDCl_3_): *δ* 7.76 (s, 8H), 6.12 (t, 8H, *J* = 5.2 Hz), 5.53 (s, 8H), 5.35 (t, 8H, *J* = 8.6 Hz), 4.76 (m, 32H), 4.57 (s, 24H), 4.44 (s, 8H), 3.67 (s, 16H), 3.61 (m, 80H), 3.52 (m, 24H), 3.46 (m, 8H), 3.37 (m, 8H), 3.16 (dd, 8H, *J* = 11.4, 4.6 Hz), 3.10 (dt, 8H, *J* = 11.1, 4.3 Hz), 2.42 (dt, 8H, *J* = 12.7, 3.1 Hz), 1.97 (m, 64H), 1.75 (m, 8H), 1.68–1.11 (m, other aliphatic ring protons), 1.66, 0.95, 0.94, 0.92, 0.80, 0.73 (s, each 24H, 48 × CH_3_), 0.87 (m, 8H), 0.66 (d, 8H, *J* = 9.2 Hz); ^13^C NMR (150 MHz, CDCl_3_): *δ* 176.18, 170.25, 169.46, 150.93, 144.88, 125.57, 109.31, 95.94, 78.85, 70.49, 70.45, 70.41, 70.15, 70.04, 69.91, 69.80, 64.43, 55.64, 55.35, 50.58, 50.09, 49.87, 46.78, 42.44, 40.72, 38.83, 38.69, 38.31, 37.72, 37.17, 34.39, 33.60, 29.65, 29.39, 29.27, 27.98, 27.39, 25.59, 20.90, 20.77, 20.64, 19.46, 18.29, 16.15, 16.12, 15.38, 14.64; MALDI-TOF MS (*m/z*) Calcd for C_408_H_641_N_32_O_96_ [M + H]^+^: 7530.74. Found 7529.87.

#### 3.3.12. Synthesis of Hexakis 6-Deoxy-6-[4-*N*-[(3,6,9,12,15,18-Hexaoxaheneicos-20-yn-1-yl)-3β-hydroxy-lup-20(29)-en-28-oyl]-aminomethyl-1*H*-1,2,3-triazol-1-yL]-2,3-dI-*O*-acetyl-α-CD (**63**)

Prepared from **37** and **48** according to general procedure B, the residue was purified by flash chromatography (eluent: CH_2_Cl_2_:CH_3_OH = 10:1) to afford **63** as a white foam with a yield of 90%. R_f_ = 0.42 (CH_2_Cl_2_:CH_3_OH = 10:1); ^1^H NMR (600 MHz, CDCl_3_): *δ* 7.68 (s, 6H), 6.12 (t, 6H, *J* = 5.0 Hz), 5.45 (t, 6H, *J* = 8.2 Hz), 5.39 (s, 6H), 4.70–4.55 (m, 48H), 3.64–3.58 (m, 120H), 3.51–3.44 (m, 24H), 3.38–3.34 (m, 6H), 3.14 (dd, 6H, *J* = 11.4, 4.6 Hz), 3.09 (dt, 6H, *J* = 11.1, 4.3 Hz), 2.41 (dt, 6H, *J* = 12.7, 3.3 Hz), 2.03–1.88 (m, 60H), 1.75 –0.84 (m, other aliphatic ring protons), 1.65, 0.93, 0.92, 0.90, 0.78, 0.72 (s, each 18H, 36 × CH_3_), 0.64 (d, 6H, *J* = 9.2 Hz); ^13^C NMR (150 MHz, CDCl_3_): *δ* 176.14, 170.22, 169.06, 150.91, 144.62, 125.58, 109.23, 96.48, 78.79, 77.00, 70.41, 70.40, 70.34, 70.10, 70.03, 69.86, 64.33, 55.59, 55.31, 50.54, 50.42, 50.04, 46.72, 42.38, 40.67, 38.78, 38.65, 38.26, 37.66, 37.12, 34.34, 33.53, 30.83, 29.34, 27.94, 27.34, 25.55, 20.85, 20.67, 20.63, 19.41, 18.24, 16.10, 16.06, 15.33, 14.59; MALDI-TOF MS (*m/z*) Calcd for C_330_H_528_N_24_NaO_84_ [M + Na]^+^: 6198.93. Found 6198.17.

#### 3.3.13. Synthesis of Heptakis 6-Deoxy-6-[4-*N*-[(3,6,9,12,15,18-Hexaoxaheneicos-20-yn-1-yl)-3β-hydroxy-lup-20(29)-en-28-oyl]-aminomethyl-1*H*-1,2,3-triazol-1-yl]-2,3-di-*O*-acetyl-β-CD (**64**)

Prepared from **37** and **49** according to general procedure B, the residue was purified by flash chromatography (eluent: CH_2_Cl_2_:CH_3_OH = 23:2) to afford **64** as a white foam with a yield of 19%. R_f_ = 0.25 (CH_2_Cl_2_:CH_3_OH = 10:1); ^1^H NMR (600 MHz, CDCl_3_): *δ* 7.75 (s, 7H), 6.13 (s, 7H), 5.49 (s, 7H), 5.35 (s, 7H), 4.84 (m, 7H), 4.72–4.66 (m, 21H), 4.57–4.47 (m, 28H), 3.62–3.60 (m, 140H), 3.53–3.46 (m, 28H), 3.39–3.36 (m, 7H), 3.16 (dd, 7H, *J* = 11.3, 4.4 Hz), 3.11 (dt, 7H, *J* = 11.1, 4.1 Hz), 2.43 (t, 7H, *J* = 10.7 Hz), 2.05–1.92 (m, 70H), 1.77–0.86 (m, other aliphatic ring protons), 1.67, 0.95, 0.94, 0.92, 0.80, 0.74 (s, each 21H, 42 × CH_3_), 0.66 (d, 7H, *J* = 9.4 Hz); ^13^C NMR (150 MHz, CDCl_3_): *δ* 176.20, 170.34, 169.37, 150.97, 144.87, 125.58, 109.30, 96.30, 78.88, 77.21, 70.65, 70.50, 70.47, 70.37, 70.17, 70.09, 69.91, 69.77, 69.67, 64.41, 55.66, 55.36, 50.60, 50.10, 49.98, 46.79, 42.45, 40.73, 38.84, 38.71, 38.32, 37.73, 37.18, 34.40, 33.60, 31.89, 30.89, 29.74, 29.66, 29.62, 29.58, 29.52, 29.48, 29.44, 29.40, 29.32, 29.28, 29.22, 27.99, 27.40, 25.61, 20.92, 20.73, 20.68, 19.47, 18.30, 16.16, 16.12, 15.38, 14.65; MALDI-TOF MS (*m/z*) Calcd for C_385_H_616_N_28_NaO_98_ [M + Na]^+^: 7228.25. Found 7228.56.

#### 3.3.14. Synthesis of Octakis 6-Deoxy-6-[4-*N*-[(3,6,9,12,15,18-Hexaoxaheneicos-20-yn-1-yl)-3β-hydroxy-lup-20(29)-en-28-oyl]-aminomethyl-1*H*-1,2,3-triazol-1-yl]-2,3-di-*O*-acetyl-γ-CD (**65**)

Prepared from **37** and **50** according to general procedure B, the residue was purified by flash chromatography (eluent: CH_2_Cl_2_:CH_3_OH = 10:1) to afford **65** as a white foam with a yield of 83%. R_f_ = 0.34 (CH_2_Cl_2_:CH_3_OH = 10:1); ^1^H NMR (600 MHz, CDCl_3_): *δ* 7.77 (s, 8H), 6.12 (s, 8H), 5.52 (s, 8H), 5.34 (t, 8H, *J* = 7.9 Hz), 4.80–4.44 (m, 64H), 3.65–3.60 (m, 160H), 3.53–3.45 (m, 32H), 3.39–3.36 (m, 8H), 3.16 (d, 8H, *J* = 10.9 Hz), 3.10 (dt, 8H, *J* = 11.2, 4.4 Hz), 2.42 (dt, 8H, *J* = 12.8, 3.4 Hz), 2.05–1.89 (m, 80H), 1.77–0.85 (m, other aliphatic ring protons), 1.66, 0.95, 0.94, 0.92, 0.80, 0.73 (s, each 24H, 48 × CH_3_), 0.66 (d, 8H, *J* = 9.3 Hz); ^13^C NMR (150 MHz, CDCl_3_): *δ* 176.19, 170.26, 169.45, 150.96, 144.84, 125.63, 109.28, 95.94, 78.85, 77.00, 75.60, 70.48, 70.45, 70.36, 70.15, 70.08, 69.88, 64.39, 55.64, 55.35, 50.59, 50.08, 46.77, 42.43, 40.72, 38.83, 38.69, 38.31, 37.72, 37.17, 34.39, 33.59, 30.88, 29.39, 27.98, 27.39, 25.59, 20.90, 20.76, 20.64, 19.46, 18.28, 16.15, 16.11, 15.37, 14.63; MALDI-TOF MS (*m/z*) Calcd for C_440_H_704_N_32_NaO_112_ [M + Na]^+^: 8257.57. Found 8257.04.

#### 3.3.15. Synthesis of Hexakis 6-Deoxy-6-[4-*N*-[(3,6,9,12,15,18,21,24-Octaoxaheptacos-26-yn-1-yl)-3β-hydroxy-lup-20(29)-en-28-oyl]-aminomethyl-1*H*-1,2,3-triazol-1-yl]-2,3-di-*O*-acetyl-α-CD (**66**)

Prepared from **38** and **48** according to general procedure B, the residue was purified by flash chromatography (eluent: CH_2_Cl_2_:CH_3_OH = 10:1) to afford **66** as a white foam with a yield of 90%. R_f_ = 0.29 (CH_2_Cl_2_:CH_3_OH = 10:1); ^1^H NMR (600 MHz, CDCl_3_): *δ* 7.69 (s, 1H), 6.09 (s, 1H), 5.47 (t, 1H, *J* = 8.6 Hz), 5.41 (s, 1H), 4.72–4.57 (m, 8H), 3.66–3.60 (m, 28H), 3.54–3.45 (m, 4H), 3.41–3.36 (m, 1H), 3.18–3.15 (m, 1H), 3.11 (dt, 1H, *J* = 11.2, 4.4 Hz), 2.43 (dt, 1H, *J* = 12.8, 3.5 Hz), 2.04–1.94 (m, 8H), 1.77–0.86 (m, other aliphatic ring protons), 1.67, 0.96, 0.95, 0.92, 0.80, 0.74 (s, each 3H, 6 × CH_3_), 0.66 (d, 1H, *J* = 9.2 Hz); ^13^C NMR (150 MHz, CDCl_3_): 176.17, 170.27, 169.07, 150.98, 144.69, 125.63, 109.29, 96.60, 78.89, 77.00, 70.53, 70.51, 70.48, 70.40, 70.18, 70.08, 69.94, 64.39, 55.66, 55.36, 50.60, 50.10, 46.79, 42.45, 40.73, 38.84, 38.70, 38.32, 37.73, 37.18, 34.40, 33.62, 30.89, 29.40, 27.98, 27.41, 25.61, 20.91, 20.73, 20.69, 19.47, 18.29, 16.16, 16.12, 15.36, 14.64; MALDI-TOF MS (*m/z*) Calcd for C_354_H_576_N_24_NaO_96_ [M + Na]^+^: 6727.56. Found 6727.62.

#### 3.3.16. Synthesis of Heptakis 6-Deoxy-6-[4-*N*-[(3,6,9,12,15,18,21,24-Octaoxaheptacos-26-yn-1-yl)-3β-hydroxy-lup-20(29)-en-28-oyl]-aminomethyl-1*H*-1,2,3-triazol-1-yl]-2,3-di-*O*-acetyl-β-CD (**67**)

Prepared from **38** and **49** according to general procedure B, the residue was purified by flash chromatography (eluent: CH_2_Cl_2_:CH_3_OH = 10:1) to afford **67** as a white foam with a yield of 64%. R_f_ = 0.34 (CH_2_Cl_2_:CH_3_OH = 10:1); ^1^H NMR (600 MHz, CDCl_3_): *δ* 7.76 (s, 1H), 6.12 (s,1 H), 5.50 (s,1 H), 5.35 (s, 1H), 4.85–4.48 (m, 8H), 3.64–3.36 (m, 33H), 3.17 (d, 1H, *J* = 10.3 Hz), 3.11 (dt, 1H, *J* = 11.2, 4.5 Hz), 2.43 (dt, 1H, *J* = 12.8, 3.5 Hz), 2.05–1.91 (m, 8H), 1.82–0.85 (m, other aliphatic ring protons), 1.67, 0.96, 0.95, 0.92, 0.81, 0.74 (s, each 3H, 6 × CH_3_), 0.67 (d, 1H, *J* = 9.2 Hz); ^13^C NMR (150 MHz, CDCl_3_): 176.20, 170.34, 169.41, 150.99, 144.83, 125.60, 109.29, 96.30, 78.90, 77.00, 70.54, 70.51, 70.49, 70.37, 70.19, 70.10, 69.94, 64.42, 55.67, 55.37, 50.61, 50.11, 46.80, 42.46, 40.74, 38.85, 38.71, 38.33, 37.74, 37.19, 34.41, 33.62, 30.90, 29.41, 28.00, 27.42, 25.62, 20.92, 19.48, 18.31, 16.17, 16.13, 15.38, 14.65. MALDI-TOF MS (*m/z*) Calcd for C_413_H_672_N_28_NaO_112_ [M + Na]^+^: 7844.99. Found 7845.54.

#### 3.3.17. Synthesis of Octakis 6-Deoxy-6-[4-*N*-[(3,6,9,12,15,18,21,24-Octaoxaheptacos-26-yn-1-yl)-3β-hydroxy-lup-20(29)-en-28-oyl]-aminomethyl-1*H*-1,2,3-triazol-1-yl]-2,3-di-*O*-acetyl-γ-CD (**68**)

Prepared from **38** and **50** according to general procedure B, the residue was purified by flash chromatography (eluent: CH_2_Cl_2_:CH_3_OH = 10:1) to afford **68** as a white foam with a yield of 73%. R_f_ = 0.34 (CH_2_Cl_2_:CH_3_OH = 10:1); ^1^H NMR (600 MHz, CDCl_3_): *δ* 7.76 (s, 1H), 6.10 (t, 1H, *J* = 8.6 Hz), 5.52 (s, 1H), 5.34 (t, 1H, *J* = 8.3 Hz), 4.80–4.44 (m, 8H), 3.66–3.60 (m, 28H), 3.53–3.45 (m, 4H), 3.40–3.35 (m, 1H), 3.16 (dd, 1H, *J* = 11.0, 3.8 Hz), 3.11 (dt, 1H, *J* = 11.2, 4.4 Hz), 2.44 (dt, 1H, *J* = 11.6, 3.5 Hz), 2.05–1.90 (m, 8H), 1.77–0.85 (m, other aliphatic ring protons), 1.67, 0.95, 0.94, 0.92, 0.80, 0.74 (s, each 3H, 6 × CH_3_), 0.66 (d, 1H, *J* = 9.1 Hz); ^13^C NMR (150 MHz, CDCl_3_): 176.16, 170.24, 169.46, 150.96, 144.87, 125.59, 109.28, 95.93, 78.87, 77.00, 75.60, 70.52, 70.50, 70.48, 70.37, 70.17, 70.06, 69.90, 69.77, 64.40, 55.64, 55.36, 50.59, 50.09, 46.78, 42.44, 40.72, 38.83, 38.70, 38.31, 37.72, 37.17, 34.39, 33.60, 30.88, 29.39, 27.98, 27.40, 25.60, 20.90, 20.77, 20.65, 19.47, 18.29, 16.15, 16.11, 15.36, 14.63; MALDI-TOF MS (*m/z*) Calcd for C_472_H_768_N_32_NaO_128_ [M + Na]^+^: 8962.42. Found 8961.29.

### 3.4. General Procedure C for the Synthesis of Multivalent BA-CD Conjugates (***69***–***86***)

The per-*O*-acetylated multivalent BA-CD conjugates (**51**–**68**) were dissolved in CH_3_OH (~5 mL per 100 mg of conjugate). CH_3_ONa (0.1 eq per mol of acetate, 30% in CH_3_OH) was added, and the solution was stirred at room temperature for 6 h. The solution was neutralized with Amberlite IR-120 H^+^ resin and filtered, the solvent was evaporated under reduced pressure, and the crude product was purified by short RP column chromatography (eluted by CH_3_OH) to afford the desired products.

#### 3.4.1. Synthesis of Hexakis 6-Deoxy-6-[4-*N*-(3β-Hydroxy-lup-20(29)-en-28-oyl)-aminomethyl-1*H*-1,2,3-triazol-1-yl]-α-CD (**69**)

Prepared from **51** according to general procedure C, the residue was purified by RP flash chromatography (eluent: methanol) to afford **69** as a white foam with a yield of 89%; ^1^H NMR (600 MHz, CDCl_3_/CD_3_OD = 1:1 *v**/**v*): *δ* 7.64 (s, 6H), 5.08 (d, 6H, *J* = 2.6 Hz), 4.65(s, 6H), 4.57 (s, 6H, overlap with H_2_O), 4.54(s, 6H), 4.38(d, 6H, *J* = 15.3 Hz), 4.33(d, 6H, *J* = 9.9 Hz), 4.23(m, 6H), 4.10(d, 6H, *J* = 15.3 Hz), 3.97(t, 6H, *J* = 9.1 Hz), 3.42(dd, 6H, *J* = 10.0, 2.5 Hz), 3.23(t, 6H, *J* = 8.8 Hz), 3.11(dd, 6H, *J* = 10.3, 5.6 Hz), 3.02(m, 6H), 2.49(t, 6H, *J* = 10.0 Hz), 2.09(d, 6H, *J* = 12.3 Hz), 2.05–0.88(m, other aliphatic ring protons), 1.63, 0.94, 0.92, 0.90, 0.81, 0.72 (s, each 18H, 36 × CH_3_), 0.66(d, 6H, *J* = 9.1 Hz); ^13^C NMR (150 MHz, CDCl_3_/CD_3_OD = 1:1 *v**/**v*): *δ* 177.97, 151.35, 145.87, 125.39, 109.94, 102.34, 83.42, 79.12, 73.67, 72.39, 70.82, 56.23, 56.06, 51.26, 50.98, 50.68, 47.24, 42.95, 41.33, 39.42, 39.36, 38.77, 38.16, 37.72, 35.06, 34.99, 33.65, 31.33, 29.9928.30, 27.39, 26.19, 21.54, 19.60, 18.91, 16.59, 16.53, 15.84; MALDI-TOF MS (*m/z*) Calcd for C_234_H_360_N_24_NaO_36_ [M + Na]^+^: 4105.70. Found 4105.12.

#### 3.4.2. Synthesis of Heptakis 6-Deoxy-6-[4-*N*-(3β-Hydroxy-lup-20(29)-en-28-oyl)-aminomethyl-1*H*-1,2,3-triazol-1-yl]-β-CD (**70**)

Prepared from **52** according to general procedure C, the residue was purified by RP flash chromatography (eluent: methanol) to afford **70** as a white foam with a yield of 85%; ^1^H NMR (400 MHz, CDCl_3_/CD_3_OD = 1:1 *v**/**v*): *δ* 7.66 (s, 7H), 5.09 (d, 7H, *J* = 2.6 Hz), 4.54 (s, 14H), 4.43–4.33 (m, 14H), 4.16–4.12 (m, 14H), 3.86 (t, 7H, *J* = 9.0 Hz), 3.47–3.43 (m, 7H), 3.25 (t, 7H, *J* = 9.1 Hz), 3.14–3.10 (m, 7H), 3.04 (s, 7H), 2.52 (t, 7H, *J* = 10.0 Hz), 2.14–2.11 (m, 7H), 1.76–0.86 (m, other aliphatic ring protons), 1.64, 0.95 (s, each 21H, 14 × CH_3_), 0.93 (s, 42H, 14 × CH_3_), 0.82, 0.73 (s, each 21H, 14 × CH_3_), 0.67 (d, 7H, *J* = 7.4 Hz); ^13^C NMR (100 MHz, CDCl_3_/CD_3_OD = 1:1 *v**/**v*): *δ* 178.29, 151.53, 146.31, 125.47, 110.10, 102.89, 83.62, 79.23, 73.57, 73.03, 71.13, 56.39, 56.27, 51.47, 51.13, 50.86, 47.43, 43.12, 41.52, 39.53, 38.96, 38.33, 37.89, 35.18, 33.81, 31.53, 30.30, 30.19, 30.03, 29.91, 28.44, 27.55, 26.39, 21.74, 19.66, 19.12, 16.75, 16.00, 15.11, 14.38; MALDI-TOF MS (*m/z*) Calcd for C_234_H_360_N_24_NaO_36_ [M + Na]^+^: 4105.70. Found 4105.12.

#### 3.4.3. Synthesis of Octakis 6-Deoxy-6-[4-*N*-(3β-Hydroxy-lup-20(29)-en-28-oyl)-aminomethyl-1*H*-1,2,3-triazol-1-yl]-γ-CD (**71**)

Prepared from **53** according to general procedure C, the residue was purified by RP flash chromatography (eluent: methanol) to afford **71** as a white foam with a yield of 85%; ^1^H NMR (600 MHz, CDCl_3_/CD_3_OD = 1:1 *v**/**v*): *δ* 7.64 (s, 8H), 5.11 (s, 8H), 4.65 (s, 8H), 4.53 (s, 8H, overlap with H_2_O), 4.41–4.14 (m, 24H), 3.85 (s, 8H), 3.46 (s, 8H), 3.24 (s, 8H), 3.11 (s, 8H), 3.03 (s, 8H), 2.51 (s, 8H), 2.11 (s, 8H), 1.76–0.94 (m, other aliphatic ring protons), 1.63, 0.94 (s, each 24H, 16 × CH_3_), 0.92 (s, 48H, 16 × CH_3_), 0.81, 0.72 (s, each 24H, 16 × CH_3_), 0.66 (s, 8H); ^13^C NMR (150 MHz, CDCl_3_/CD_3_OD = 1:1 *v**/**v*): *δ* 177.89, 151.36, 145.97, 125.18, 109.90, 102.51, 82.92, 79.09, 73.08, 70.71, 63.77, 56.19, 56.02, 51.23, 50.66, 47.18, 42.93, 41.30, 39.39, 39.32, 38.74, 38.10, 37.69, 35.12, 34.97, 33.64, 31.32, 30.10, 29.96, 28.29, 27.36, 26.17, 21.52, 19.60, 18.88, 16.58, 16.53, 15.82, 14.99; MALDI-TOF MS (*m/z*) Calcd for C_312_H_480_N_32_NaO_48_ [M + Na]^+^: 5466.60. Found 5467.72.

#### 3.4.4. Synthesis of Hexakis 6-Deoxy-6-[4-*N*-[(2-(2-Propyn-1-yloxy)ethyl)-3β-hydroxy-lup-20(29)-en-28-oyl]-aminomethyl-1*H*-1,2,3-triazol-1-yl]-α-CD (**72**)

Prepared from **54** according to general procedure C, the residue was purified by RP flash chromatography (eluent: methanol) to afford **72** as a white foam with a yield of 88%; ^1^H NMR (600 MHz, CDCl_3_/CD_3_OD = 1:1 *v**/**v*): *δ* 7.86 (s, 6H), 5.10 (s, 6H), 4.68–4.45 (m, 36H, overlap with H_2_O), 4.23 (s, 6H), 3.99 (m, 6H), 3.50–3.41 (m, 18H), 3.24 (m, 6H), 3.12 (dd, 6H, *J* = 10.9, 5.2 Hz), 3.05 (m, 6H), 2.49 (t, 6H, *J* = 11.1 Hz), 2.09 (d, 6H, *J* = 12.5 Hz), 1.86–1.77 (m, 12H), 1.65–0.85 (m, other aliphatic ring protons), 1.65, 0.95 (s, each 18H, 12 × CH_3_), 0.92 (s, 36H, 12 × CH_3_), 0.81, 0.72 (s, each 18H, 12 × CH_3_), 0.66 (d, 6H, *J* = 9.9 Hz); ^13^C NMR (150 MHz, CDCl_3_/CD_3_OD = 1:1 *v**/**v*): *δ* 178.33, 151.56, 145.31, 126.41, 109.92, 102.68, 83.53, 79.18, 73.79, 72.57, 70.96, 70.11, 64.44, 58.01, 56.42, 56.22, 51.40, 50.83, 49.85, 47.48, 43.06, 41.46, 39.56, 39.48, 38.97, 38.35, 37.84, 35.13, 33.80, 31.50, 30.10, 28.41, 27.51, 26.34, 21.67, 19.70, 19.02, 16.64, 15.94, 15.11; MALDI-TOF MS (*m/z*) Calcd for C_246_H_384_N_24_NaO_42_ [M + Na]^+^: 4369.85. Found 4370.96.

#### 3.4.5. Synthesis of Heptakis 6-Deoxy-6-[4-*N*-[(2-(2-Propyn-1-yloxy)ethyl)-3β-hydroxy-lup-20(29)-en-28-oyl]-aminomethyl-1*H*-1,2,3-triazol-1-yl]-β-CD (**73**)

Prepared from **55** according to general procedure C, the residue was purified by RP flash chromatography (eluent: methanol) to afford **73** as a white foam with a yield of 85%; ^1^H NMR (600 MHz, CDCl_3_/CD_3_OD = 1:1 *v**/**v*): *δ* 7.86 (s, 7H), 5.12 (d, 7H, *J* = 3.0 Hz), 4.68 (s, 7H), 4.58–4.44 (m, 35H), 4.17 (m, 7H), 3.88 (t, 7H, *J* = 9.2 Hz), 3.54–3.49 (m, 14H), 3.44–3.42 (m, 7H), 3.23 (t, 7H, *J* = 9.1 Hz), 3.12 (dd, 7H, *J* = 11.0, 5.2 Hz), 3.05 (dt, 7H, *J* = 11.2, 4.4 Hz), 2.50 (t, 7H, *J* = 12.7 Hz), 2.09 (d, 7H, *J* = 12.4 Hz), 1.85–1.77 (m, 14H), 1.65–0.85 (m, other aliphatic ring protons), 1.65, 0.95 (s, each 21H, 14 × CH_3_), 0.93 (s, 42H, 14 × CH_3_), 0.82, 0.73 (s, each 21H, 14 × CH_3_), 0.67 (d, 7H, *J* = 9.6 Hz); ^13^C NMR (150 MHz, CDCl_3_/CD_3_OD = 1:1 *v/v*): *δ* 178.39, 151.61, 145.35, 126.41, 109.96, 102.96, 83.50, 79.22, 73.56, 73.03, 70.96, 70.12, 64.51, 58.03, 56.47, 56.29, 51.47, 50.90, 49.86, 47.53, 43.11, 41.52, 39.62, 39.53, 39.03, 38.39, 37.89, 35.20, 33.84, 31.56, 30.16, 28.46, 27.56, 26.40, 21.72, 19.73, 19.08, 16.69, 15.98, 15.14; MALDI-TOF MS (*m/z*) Calcd for C_287_H_448_N_28_NaO_49_ [M + Na]^+^: 5097.88. Found 5097.29.

#### 3.4.6. Synthesis of Octakis 6-Deoxy-6-[4-*N*-[(2-(2-Propyn-1-yloxy)ethyl)-3β-hydroxy-lup-20(29)-en-28-oyl]-aminomethyl-1*H*-1,2,3-triazol-1-yl]-γ-CD (**74**)

Prepared from **56** according to general procedure C, the residue was purified by RP flash chromatography (eluent: methanol) to afford **74** as a white foam with a yield of 86%; ^1^H NMR (600 MHz, CDCl_3_/CD_3_OD = 1:1 *v**/**v*): *δ* 7.84 (s, 8H), 5.16 (d, 8H, *J* = 2.8 Hz), 4.68–4.49 (m, 48H), 4.16 (m, 8H), 3.88 (t, 8H, *J* = 9.2 Hz), 3.53–3.41 (m, 24H), 3.20 (t, 8H, *J* = 9.2 Hz), 3.12–3.09 (m, 8H), 3.04 (dt, 8H, *J* = 11.1, 4.5 Hz), 2.46 (t, 8H, *J* = 9.9 Hz), 2.05 (d, 8H, *J* = 12.5 Hz), 1.85–1.75 (m, 16H), 1.63–0.84 (m, other aliphatic ring protons), 1.63, 0.93, 0.91, 0.90, 0.79, 0.71 (s, each 24H, 48 × CH_3_), 0.64 (d, 8H, *J* = 9.6 Hz); ^13^C NMR (150 MHz, CDCl_3_/CD_3_OD = 1:1 *v**/**v*): *δ* 177.90, 151.36, 145.09, 126.08, 109.76, 102.39, 82.44, 79.03, 73.13, 72.96, 70.47, 69.95, 64.37, 57.90, 56.17, 55.94, 51.13, 50.58, 49.86, 47.21, 42.85, 41.21, 39.30, 39.26, 38.77, 38.10, 37.61, 34.90, 33.67, 31.28, 29.87, 28.26, 27.30, 26.10, 21.42, 19.63, 18.78, 16.49, 16.45, 15.76, 14.98; MALDI-TOF MS (*m/z*) Calcd for C_328_H_512_N_32_NaO_56_ [M + Na]^+^: 5822.86. Found 5822.75.

#### 3.4.7. Synthesis of Hexakis 6-Deoxy-6-[4-*N*-[(2-(2-(2-Propyn-1-yloxy)ethoxy)ethyl)-3β-hydroxy-lup-20(29)-en-28-oyl]-aminomethyl-1*H*-1,2,3-triazol-1-yl]-α-CD (**75**)

Prepared from **57** according to general procedure C, the residue was purified by RP flash chromatography (eluent: methanol) to afford **75** as a white foam with a yield of 91%; ^1^H NMR (600 MHz, CDCl_3_/CD_3_OD = 1:1 *v**/**v*): *δ* 7.92 (s, 6H), 5.12 (d, 6H, *J* = 2.7 Hz), 4.68–4.47 (m, 36H, overlap with H_2_O), 4.25 (m, 6H), 3.99 (t, 6H, *J* = 9.1 Hz), 3.62–3.56 (m, 24H), 3.50–3.45 (m, 12H), 3.42–3.40 (m, 6H), 3.24 (t, 6H, *J* = 8.9 Hz), 3.12 (dd, 6H, *J* = 11.0, 5.2 Hz), 3.06 (dt, 6H, *J* = 10.9, 4.3 Hz), 2.48 (dt, 6H, *J* = 12.5, 3.2 Hz), 2.08 (d, 6H, *J* = 13.0 Hz), 1.88–1.77 (m, 12H), 1.66–0.85 (m, other aliphatic ring protons), 1.66, 0.96, 0.93, 0.92, 0.81, 0.72 (s, each 18H, 36 × CH_3_), 0.67 (d, 6H, *J* = 10.0 Hz); ^13^C NMR (150 MHz, CDCl_3_/CD_3_OD = 1:1 *v**/**v*): *δ* 178.29, 151.58, 145.07, 126.76, 109.89, 102.56, 83.42, 79.19, 73.86, 72.58, 70.98, 70.75, 70.50, 64.66, 56.44, 56.23, 54.14, 51.41, 50.84, 47.52, 43.08, 41.46, 39.56, 39.49, 38.96, 38.39, 37.84, 35.14, 33.84, 31.51, 30.11, 28.42, 27.52, 26.36, 21.65, 19.72, 19.02, 18.16, 16.63, 15.93, 15.12; MALDI-TOF MS (*m/z*) Calcd for C_258_H_408_N_24_NaO_48_ [M + Na]^+^: 4637.21. Found 4636.52.

#### 3.4.8. Synthesis of Heptakis 6-Deoxy-6-[4-*N*-[(2-(2-(2-Propyn-1-yloxy)ethoxy)ethyl)-3β-hydroxy-lup-20(29)-en-28-oyl]-aminomethyl-1*H*-1,2,3-triazol-1-yl]-β-CD (**76**)

Prepared from **58** according to general procedure C, the residue was purified by RP flash chromatography (eluent: methanol) to afford **76** as a white foam with a yield of 82%; ^1^H NMR (600 MHz, CDCl_3_/CD_3_OD = 1:1 *v**/**v*): *δ* 7.93 (s, 7H), 5.13 (d, 7H, *J* = 2.8 Hz), 4.70–4.44 (m, 42H, overlap with H_2_O), 4.18 (m, 7H), 3.87 (t, 7H, *J* = 9.1 Hz), 3.63–3.56 (m, 28H), 3.50–3.42 (m, 21H), 3.23 (t, 7H, *J* = 10.4 Hz), 3.12 (dd, 7H, *J* = 11.0, 5.2 Hz), 3.06 (dt, 7H, *J* = 10.9, 4.1 Hz), 2.49 (t, 7H, *J* = 11.6 Hz), 2.09 (d, 7H, *J* = 12.8 Hz), 1.88–1.78 (m, 14H), 1.66–0.82 (m, other aliphatic ring protons), 1.66, 0.96, 0.93, 0.92, 0.82, 0.73 (s, each 21H, 42 × CH_3_), 0.67 (d, 7H, *J* = 10.1 Hz); ^13^C NMR (150 MHz, CDCl_3_/CD_3_OD = 1:1 *v**/**v*): *δ* 178.33, 151.61, 145.06, 126.80, 109.92, 102.87, 83.39, 79.21, 73.62, 73.05, 70.97, 70.79, 70.52, 64.70, 58.02, 56.46, 56.27, 51.45, 50.88, 49.86, 47.54, 43.11, 41.50, 39.60, 39.52, 39.48, 39.00, 38.41, 37.88, 35.19, 33.86, 31.55, 30.14, 28.45, 27.55, 26.40, 21.69, 19.74, 19.05, 18.18, 16.66, 15.96, 15.14; MALDI-TOF MS (*m/z*) Calcd for C_301_H_476_N_28_NaO_56_ [M + Na]^+^: 5402.52. Found 5402.60.

#### 3.4.9. Synthesis of Octakis 6-Deoxy-6-[4-*N*-[(2-(2-(2-Propyn-1-yloxy)ethoxy)ethyl)-3β-hydroxy-lup-20(29)-en-28-oyl]-aminomethyl-1*H*-1,2,3-triazol-1-yl]-γ-CD (**77**)

Prepared from **59** according to general procedure C, the residue was purified by RP flash chromatography (eluent: methanol) to afford **77** as a white foam with a yield of 83%; ^1^H NMR (600 MHz, CDCl_3_/CD_3_OD = 1:1 *v**/**v*): *δ* 7.93 (s, 8H), 5.18 (d, 8H, *J* = 2.8 Hz), 4.68–4.45 (m, 48H, overlap with H_2_O), 4.19 (m, 8H), 3.89 (t, 8H, *J* = 9.2 Hz), 3.63–3.58 (m, 32H), 3.52–3.43 (m, 24H), 3.25–3.22 (m, 8H), 3.12 (dd, 8H, *J* = 10.9, 5.2 Hz), 3.06 (dt, 8H, *J* = 10.9, 4.3 Hz), 2.49 (dt, 8H, *J* = 12.6, 3.0 Hz), 2.09 (d, 8H, *J* = 12.8 Hz), 1.90–1.77 (m, 16H), 1.65–0.86 (m, other aliphatic ring protons), 1.65, 0.96 (s, each 24H, 16 × CH_3_), 0.93 (s, 48H, 16 × CH_3_), 0.82, 0.73 (s, each 24H, 16 × CH_3_), 0.67 (d, 8H, *J* = 10.0 Hz); ^13^C NMR (150 MHz, CDCl_3_/CD_3_OD = 1:1 *v**/**v*): *δ* 178.28, 151.58, 145.04, 126.77, 109.91, 102.66, 82.87, 79.19, 73.45, 73.28, 70.76, 70.50, 64.72, 58.01, 56.44, 56.25, 51.42, 50.85, 49.86, 47.51, 43.09, 41.47, 39.58, 39.50, 39.47, 38.98, 38.39, 37.86, 35.17, 33.85, 31.52, 30.12, 28.44, 27.52, 26.37, 21.68, 19.74, 19.03, 18.17, 16.65, 15.95, 15.14; MALDI-TOF MS (*m/z*) Calcd for C_344_H_544_N_32_NaO_64_ [M + Na]^+^: 6175.29. Found 6175.66.

#### 3.4.10. Synthesis of Hexakis 6-Deoxy-6-[4-*N*-[(3,6,9,12-Tetraoxapentadec-14-yn-1-yl)-3β-hydroxy-lup-20(29)-en-28-oyl]-aminomethyl-1*H*-1,2,3-triazol-1-yl]-α-CD (**78**)

Prepared from **60** according to general procedure C, the residue was purified by RP flash chromatography (eluent: methanol) to afford **78** as a white foam with a yield of 92%; ^1^H NMR (600 MHz, CDCl_3_/CD_3_OD = 1:1 *v**/**v*): *δ* 7.94 (s, 6H), 5.14 (s, 6H), 4.71 (s, 6H), 4.58–4.51 (m, 18H), 4.27 (s, 6H), 4.01 (t, 6H, *J* = 9.0 Hz), 3.63–3.52 (m, 90H), 3.44–3.25 (m, 30H), 3.14 (dd, 6H, *J* = 10.9, 5.3 Hz), 3.08 (dt, 6H, *J* = 11.0, 4.3 Hz), 2.49 (t, 6H, *J* = 11.8 Hz), 2.09 (d, 6H, *J* = 13.0 Hz), 1.90 (m, 6H), 1.81 (m, 6H), 1.68–0.88 (m, other aliphatic ring protons), 1.68, 0.98 (s, each 18H, 12 × CH_3_), 0.95 (s, 36H, 12 × CH_3_), 0.84, 0.75 (s, each 18H, 12 × CH_3_), 0.69 (d, 6H, *J* = 10.2 Hz); ^13^C NMR (150 MHz, CDCl_3_/CD_3_OD = 1:1 *v**/**v*): *δ* 178.22, 151.55, 145.01, 126.74, 109.85, 102.57, 83.42, 79.16, 73.81, 72.54, 71.04, 71.01, 70.71, 70.51, 70.48, 64.56, 56.42, 56.19, 51.37, 51.12, 50.80, 47.51, 43.05, 41.43, 39.53, 39.52, 39.46, 38.90, 38.40, 37.81, 35.10, 33.80, 31.47, 30.20, 30.06, 29.83, 28.39, 27.70, 27.50, 26.33, 23.22, 21.62, 19.70, 18.97, 16.60, 15.90, 15.10, 14.33; MALDI-TOF MS (*m/z*) Calcd for C_282_H_456_N_24_NaO_60_ [M + Na]^+^: 5165.85. Found 5165.20.

#### 3.4.11. Synthesis of Heptakis 6-Deoxy-6-[4-*N*-[(3,6,9,12-Tetraoxapentadec-14-yn-1-yl)-3β-hydroxy-lup-20(29)-en-28-oyl]-aminomethyl-1*H*-1,2,3-triazol-1-yl]-β-CD (**79**)

Prepared from **61** according to general procedure C, the residue was purified by RP flash chromatography (eluent: methanol) to afford **79** as a white foam with a yield of 88%; ^1^H NMR (600 MHz, CDCl_3_/CD_3_OD = 1:1 *v**/**v*): *δ* 7.96 (s, 7H), 5.15 (s, 7H), 4.72 (s, 7H), 4.59–4.53 (m, 21H), 4.18 (s, 7H), 3.89 (m, 7H), 3.64–3.34 (m, 133H), 3.26 (t, 7H, *J* = 9.0 Hz), 3.15 (dd, 7H, *J* = 10.8, 5.4 Hz), 3.09 (dt, 7H, *J* = 11.0, 4.1 Hz), 2.49 (t, 7H, *J* = 11.6 Hz), 2.09 (d, 7H, *J* = 13.1 Hz), 1.91 (m, 7H), 1.81 (m, 7H), 1.69–0.89 (m, other aliphatic ring protons), 1.69, 0.99 (each 21H, 14 × CH_3_), 0.96 (s, 42H, 14 × CH_3_), 0.84, 0.76 (s, each 21H, 14 × CH_3_), 0.70 (d, 7H, *J* = 10.3 Hz); ^13^C NMR (150 MHz, CDCl_3_/CD_3_OD = 1:1 *v**/**v*): *δ* 178.17, 151.52, 144.87, 126.81, 109.85, 102.72, 83.35, 79.15, 73.48, 72.87, 71.01, 70.99, 70.68, 70.56, 70.46, 64.47, 56.39, 56.16, 51.33, 51.13, 50.77, 47.49, 43.03, 41.40, 39.50, 39.48, 39.43, 38.89, 38.38, 37.79, 35.08, 33.79, 31.45, 30.17, 30.04, 29.81, 28.38, 27.68, 27.47, 26.30, 23.19, 21.59, 19.69, 18.95, 16.59, 15.89, 15.09, 14.31; MALDI-TOF MS (*m/z*) Calcd for C_329_H_533_N_28_O_70_ [M + H]^+^: 6001.01. Found 6001.28.

#### 3.4.12. Synthesis of Octakis 6-Deoxy-6-[4-*N*-[(3,6,9,12-Tetraoxapentadec-14-yn-1-yl)-3β-hydroxy-lup-20(29)-en-28-oyl]-aminomethyl-1*H*-1,2,3-triazol-1-yl]-γ-CD (**80**)

Prepared from **62** according to general procedure C, the residue was purified by RP flash chromatography (eluent: methanol) to afford **80** as a white foam with a yield of 94%; ^1^H NMR (600 MHz, CDCl_3_/CD_3_OD = 1:1 *v**/**v*): *δ* 7.94 (s, 8H), 5.21 (s, 8H), 4.72 (s, 8H), 4.21 (s, 8H), 3.92 (s, 8H), 3.64–3.33 (m, 144H), 3.26 (s, 8H), 3.15 (dd, 8H, *J* = 10.4, 5.8 Hz), 3.09 (dt, 8H, *J* = 11.0, 4.3 Hz), 2.48 (m, 8H), 2.08 (d, 8H, *J* = 13.0 Hz), 1.91 (m, 8H), 1.81 (m, 8H), 1.69–0.89 (m, other aliphatic ring protons), 1.69, 0.99, 0.95, 0.94, 0.84, 0.76 (each 24H, 48 × CH_3_), 0.69 (d, 8H, *J* = 10.0 Hz); ^13^C NMR (150 MHz, CDCl_3_/CD_3_OD = 1:1 *v**/**v*): *δ* 177.96, 151.41, 109.76, 102.39, 82.57, 79.06, 73.22, 73.04, 70.91, 70.89, 70.57, 70.40, 64.52, 56.27, 56.02, 51.20, 50.64, 47.35, 42.93, 41.28, 39.38, 39.33, 38.78, 38.26, 37.68, 34.96, 33.71, 31.34, 30.07, 29.92, 29.71, 28.31, 27.59, 27.38, 26.18, 23.10, 21.48, 19.66, 18.84, 16.51, 16.50, 15.81, 15.03, 14.29; MALDI-TOF MS (*m/z*) Calcd for C_376_H_608_N_32_NaO_80_ [M + Na]^+^: 6880.13. Found 6880.93.

#### 3.4.13. Synthesis of Hexakis 6-Deoxy-6-[4-*N*-[(3,6,9,12,15,18-Hexaoxaheneicos-20-yn-1-yl)-3β-hydroxy-lup-20(29)-en-28-oyl]-aminomethyl-1*H*-1,2,3-triazol-1-yl]-α-CD (**81**)

Prepared from **63** according to general procedure C, the residue was purified by RP flash chromatography (eluent: methanol) to afford **81** as a white foam with a yield of 90%; ^1^H NMR (600 MHz, CDCl_3_/CD_3_OD = 1:1 *v**/**v*): *δ* 7.92 (s, 6H), 5.13 (s, 6H), 4.60–4.45 (m, 30H), 4.26 (s, 6H), 3.98 (t, 6H, *J* = 8.9 Hz), 3.63–3.38 (m, 144H), 3.29–3.22 (m, 6H), 3.12 (dd, 6H, *J* = 10.3, 5.8 Hz), 3.06 (dt, 6H, *J* = 13.5, 6.7 Hz), 2.46 (dt, 6H, *J* = 12.6, 3.2 Hz), 2.08–2.05 (m, 6H), 1.93–1.76 (m, 12H), 1.66–0.86 (m, other aliphatic ring protons), 1.66, 0.97 (s, each 18H, 12 × CH_3_), 0.93 (s, 36H, 12 × CH_3_), 0.82, 0.73 (s, each 18H, 12 × CH_3_), 0.67 (d, 6H, *J* = 9.2 Hz); ^13^C NMR (150 MHz, CDCl_3_/CD_3_OD = 1:1 *v**/**v*): *δ* 178.32, 151.61, 145.12, 126.78, 109.85, 102.58, 83.52, 79.21, 73.89, 72.59, 71.12, 71.09, 71.04, 70.76, 70.53, 64.63, 56.48, 56.26, 51.44, 51.12, 50.87, 49.85, 47.57, 43.10, 41.49, 39.59, 39.51, 38.94, 38.46, 37.87, 35.15, 33.82, 31.52, 30.11, 28.42, 27.55, 26.39, 21.67, 19.72, 19.02, 18.18, 16.62, 15.93, 15.11; MALDI-TOF MS (*m/z*) Calcd for C_306_H_504_N_24_NaO_72_ [M + Na]^+^: 5694.48. Found 5694.02.

#### 3.4.14. Synthesis of Heptakis 6-Deoxy-6-[4-*N*-[(3,6,9,12,15,18-Hexaoxaheneicos-20-yn-1-yl)-3β-hydroxy-lup-20(29)-en-28-oyl]-aminomethyl-1*H*-1,2,3-triazol-1-yl]-β-CD (**82**)

Prepared from **64** according to general procedure C, the residue was purified by RP flash chromatography (eluent: methanol) to afford **82** as a white foam with a yield of 85%; ^1^H NMR (600 MHz, CDCl_3_/CD_3_OD = 1:1 *v**/**v*): *δ* 7.94 (s, 7H), 5.12 (s, 7H), 4.69 (s, 7H), 4.56–4.39 (m, 35H), 4.15 (s, 7H), 3.86 (t, 7H, *J* = 9.0 Hz), 3.63–3.39 (m, 168H), 3.29–3.25 (m, 7H), 3.12 (dd, 7H, *J* = 10.3, 5.8 Hz), 3.06 (dt, 7H, *J* = 13.4, 6.6 Hz), 2.47 (t, 7H, *J* = 11.4 Hz), 2.08–2.05 (m, 7H), 1.90–1.76 (m, 14H), 1.66–0.86 (m, other aliphatic ring protons), 1.66, 0.97 (s, each 21H, 14 × CH_3_), 0.93 (s, 42H, 14 × CH_3_), 0.82, 0.73 (s, each 21H, 14 × CH_3_), 0.67 (d, 7H, *J* = 9.3 Hz); ^13^C NMR (150 MHz, CDCl_3_/CD_3_OD = 1:1 *v**/**v*): *δ* 178.33, 151.61, 144.97, 126.92, 109.87, 102.82, 83.53, 79.21, 73.60, 72.98, 71.12, 71.10, 71.06, 70.77, 70.63, 70.53, 64.53, 56.48, 56.27, 51.45, 51.20, 50.87, 49.85, 47.58, 43.11, 41.50, 39.59, 39.51, 38.95, 38.46, 37.87, 35.16, 33.83, 31.53, 30.12, 28.43, 27.56, 26.40, 21.68, 19.72, 19.03, 18.18, 16.63, 15.94, 15.12; MALDI-TOF MS (*m/z*) Calcd for C_357_H_588_N_28_NaO_84_ [M + Na]^+^: 6639.73. Found 6639.63.

#### 3.4.15. Synthesis of Octakis 6-Deoxy-6-[4-*N*-[(3,6,9,12,15,18-Hexaoxaheneicos-20-yn-1-yl)-3β-hydroxy-lup-20(29)-en-28-oyl]-aminomethyl-1*H*-1,2,3-triazol-1-yl]-γ-CD (**83**)

Prepared from **65** according to general procedure C, the residue was purified by RP flash chromatography (eluent: methanol) to afford **83** as a white foam with a yield of 89%; ^1^H NMR (400 MHz, CDCl_3_/CD_3_OD = 1:1 *v**/**v*): *δ* 7.93 (s, 8H), 5.18 (s, 8H), 4.56–4.48 (m, 40H), 4.17 (s, 8H), 3.88 (t, 8H, *J* = 9.0 Hz), 3.63–3.38 (m, 192H), 3.29–3.25 (m, 8H), 3.12 (dd, 8H, *J* = 10.3, 5.9 Hz), 3.06 (dt, 8H, *J* = 13.4, 6.4 Hz), 2.47 (t, 8H, *J* = 11.4 Hz), 2.08–2.05 (m, 8H), 1.90–0.86 (m, other aliphatic ring protons), 1.66, 0.97 (s, each 24H, 16 × CH_3_), 0.93 (s, 48H, 16 × CH_3_), 0.82, 0.73 (s, each 24H, 16 × CH_3_), 0.67 (d, 8H, *J* = 9.3 Hz); ^13^C NMR (150 MHz, CDCl_3_/CD_3_OD = 1:1 *v**/**v*): *δ* 178.33, 151.62, 145.08, 126.85, 109.87, 102.69, 83.05, 79.22, 73.47, 73.29, 71.13, 71.11, 71.07, 70.88, 70.77, 70.57, 70.53, 64.66, 56.49, 56.28, 51.46, 50.88, 49.86, 47.58, 43.11, 41.50, 39.60, 39.53, 38.95, 38.47, 37.88, 35.17, 33.83, 31.54, 30.13, 28.45, 27.57, 26.41, 21.69, 19.73, 19.04, 18.18, 16.64, 15.95, 15.13; MALDI-TOF MS (*m/z*) Calcd for C_408_H_672_N_32_NaO_96_ [M + Na]^+^: 7584.98. Found 7584.17.

#### 3.4.16. Synthesis of Hexakis 6-Deoxy-6-[4-*N*-[(3,6,9,12,15,18,21,24-Octaoxaheptacos-26-yn-1-yl)-3β-hydroxy-lup-20(29)-en-28-oyl]-aminomethyl-1*H*-1,2,3-triazol-1-yl]-α-CD (**84**)

Prepared from **66** according to general procedure C, the residue was purified by RP flash chromatography (eluent: methanol) to afford **84** as a white foam with a yield of 90%; ^1^H NMR (600 MHz, CDCl_3_/CD_3_OD = 1:1 *v**/**v*): *δ* 7.92 (s, 1H), 5.11 (s, 1H), 4.69 (m, 1H), 4.56–4.48 (m, 4H), 4.38 (s, 1H), 4.22 (s, 1H), 3.98 (t, 1H, *J* = 8.9 Hz), 3.63–3.59 (m, 28H), 3.51 (t, 2H, *J* = 5.3 Hz), 3.45–3.39 (m, 2H), 3.25 (m, 1H), 3.12 (dd, 1H, *J* = 10.9, 5.4 Hz), 3.05 (dt, 1H, *J* = 11.0, 4.3 Hz), 2.45 (dt, 1H, *J* = 13.0, 3.1 Hz), 2.06 (m, 1H), 1.92–1.84 (m, 1H), 1.80–1.76 (m, 1H), 1.68–0.85 (m, other aliphatic ring protons), 1.66, 0.96, 0.93, 0.92, 0.81, 0.72 (s, each 3H, 6 × CH_3_), 0.66 (d, 1H, *J* = 10.0 Hz); ^13^C NMR (150 MHz, CDCl_3_/CD_3_OD = 1:1 *v**/**v*): 178.15, 151.52, 144.83, 126.82, 109.79, 102.51, 83.55, 79.14, 73.75, 72.44, 71.03, 71.02, 71.00, 70.97, 70.67, 70.54, 70.47, 64.43, 56.38, 56.14, 51.32, 50.75, 47.47, 43.01, 41.38, 39.49, 39.41, 38.85, 38.37, 37.77, 35.04, 33.76, 31.43, 30.01, 28.35, 27.46, 26.28, 21.57, 19.68, 18.92, 16.54, 15.85, 15.06; MALDI-TOF MS (*m/z*) Calcd for C_330_H_552_N_24_NaO_84_ [M + Na]^+^: 6223.12. Found 6223.38.

#### 3.4.17. Synthesis of Heptakis 6-Deoxy-6-[4-*N*-[(3,6,9,12,15,18,21,24-Octaoxaheptacos-26-yn-1-yl)-3β-hydroxy-lup-20(29)-en-28-oyl]-aminomethyl-1*H*-1,2,3-triazol-1-yl]-β-CD (**85**)

Prepared from **67** according to general procedure C, the residue was purified by RP flash chromatography (eluent: methanol) to afford **85** as a white foam with a yield of 87%; ^1^H NMR (600 MHz, CDCl_3_/CD_3_OD = 1:1 *v**/**v*): *δ* 7.93 (s, 1H), 5.11 (s, 1H), 4.69 (m, 1H), 4.56–4.49 (m, 4H), 4.35 (m, 1H), 4.13 (s, 1H), 3.84 (t, 1H, *J* = 8.9 Hz), 3.63–3.59 (m, 28H), 3.51 (t, 2H, *J* = 5.2 Hz), 3.47–3.39 (m, 2H), 3.24 (m, 1H), 3.12 (dd, 1H, *J* = 10.9, 5.3 Hz), 3.05 (dt, 1H, *J* = 11.0, 4.4 Hz), 2.45 (dt, 1H, *J* = 12.8, 3.2 Hz), 2.06 (m, 1H), 1.92–1.84 (m, 1H), 1.80–1.76 (m, 1H), 1.66–0.85 (m, other aliphatic ring protons), 1.66, 0.96, 0.93, 0.92, 0.81, 0.72 (s, each 3H, 6 × CH_3_), 0.66 (d, 1H, *J* = 10.0 Hz); ^13^C NMR (150 MHz, CDCl_3_/CD_3_OD = 1:1 *v**/**v*): 178.14, 151.51, 144.79, 126.84, 109.79, 102.68, 83.40, 79.14, 73.45, 72.83, 71.01, 70.94, 70.66, 70.57, 70.47, 64.38, 56.38, 56.13, 51.31, 50.74, 47.47, 43.01, 41.38, 39.48, 39.41, 38.85, 38.37, 37.77, 35.04, 33.76, 31.42, 30.01, 28.35, 27.46, 26.28, 21.56, 19.68, 18.92, 16.54, 15.85, 15.06; MALDI-TOF MS (*m/z*) Calcd for C_385_H_644_N_28_NaO_98_ [M + Na]^+^: 7256.47. Found 7254.72.

#### 3.4.18. Synthesis of Octakis 6-Deoxy-6-[4-*N*-[(3,6,9,12,15,18,21,24-Octaoxaheptacos-26-yn-1-yl)-3β-hydroxy-lup-20(29)-en-28-oyl]-aminomethyl-1*H*-1,2,3-triazol-1-yl]-γ-CD (**86**)

Prepared from **68** according to general procedure C, the residue was purified by RP flash chromatography (eluent: methanol) to afford **86** as a white foam with a yield of 92%; ^1^H NMR (600 MHz, CDCl_3_/CD_3_OD = 1:1 *v**/**v*): *δ* 7.90 (s, 1H), 5.16 (s, 1H), 4.68 (s, 1H), 4.55 (m, 5H), 4.16 (s, 1H), 3.87 (s, 1H), 3.63–3.59 (m, 28H), 3.51 (t, 2H, *J* = 5.2 Hz), 3.45–3.38 (m, 2H), 3.22 (s, 1H), 3.11 (dd, 1H, *J* = 10.2, 6.1 Hz), 3.05 (dt, 1H, *J* = 11.0, 4.3 Hz), 2.42 (dt, 1H, *J* = 12.9, 3.1 Hz), 2.03 (m, 1H), 1.91–1.84 (m, 1H), 1.78–1.75 (m, 1H), 1.65–0.85 (m, other aliphatic ring protons), 1.65, 0.95, 0.92, 0.91, 0.80 (s, each 3H, 6 × CH_3_), 0.65 (d, 1H, *J* = 10.0 Hz); ^13^C NMR (150 MHz, CDCl_3_/CD_3_OD = 1:1 *v**/**v*): 177.84, 151.36, 144.81, 126.57, 109.68, 102.30, 82.51, 79.03, 73.13, 72.93, 70.88, 70.84, 70.81, 70.49, 70.38, 70.33, 64.43, 56.21, 55.93, 51.11, 50.55, 47.29, 42.86, 41.20, 39.31, 39.29, 39.25, 38.70, 38.21, 37.60, 34.86, 33.65, 31.26, 29.83, 28.24, 27.32, 26.10, 21.39, 19.62, 18.75, 16.44, 16.42, 15.73, 14.97. MALDI-TOF MS (*m/z*) Calcd for C_440_H_736_N_32_NaO_112_ [M + Na]^+^: 8289.83. Found 8289.80.

### 3.5. Cytotoxicity Test

The cytotoxicity of the synthesized BA-CD conjugates was evaluated with the CellTiter-Glo luminescent cell viability assay kit. Briefly, 10,000 MDCK cells in DMEM supplemented with 1% FBS were grown in 96-well plates and incubated at 37 °C. After 24 h, the cells were treated or mock-treated with 100 µM test compounds and collected at 36 h. An equal volume of CellTiter-Glo reagent was added to the cells and mixed for 2 min on an orbital shaker. After stabilization at room temperature for 10 min, the luminescence intensity was measured by an Infinite M2000 PRO™ instrument (Tecan Group Ltd., Männedorf, Switzerland).

### 3.6. CPE Reduction Assay

MDCK cells were seeded at 1.0 × 10^4^ cells per well in 96-well plates and cultured overnight. When the cells had grown to approximately 70-80% confluence, the media were removed, and the cells were infected with influenza A/WSN/33 virus at a multiplicity of infection (MOI) of 0.2 in DMEM (with 1% FBS and 2 µg/mL TPCK-treated trypsin) containing the corresponding concentration of test samples with two-fold serial dilution (0.78, 1.56, 3.13, 6.25, 12.50, 25.00, 50.00, and 100.00 µM) from the stock solutions (2.0 mM). All plates were incubated at 37 °C and 5% CO_2_ for 36 h, and cell viability was determined using CellTiter-Glo reagent, as described above.

### 3.7. Statistical Analysis

Data was analyzed using GraphPad Prism 8.3.0 (GraphPad Software Inc., San Diego, CA, USA). The half maximal inhibitory concentration (IC_50_) and cytotoxicity concentration for 50% cell death (CC_50_) was calculated using a non-linear regression dose response curve. Selectivity Index (SI) was calculated as the rate of CC_50_ to IC_50_.

## 4. Conclusions

In summary, an efficient and conventional method is presented here for the synthesis of multivalent BA derivatives by using α-, β- and γ-CD as scaffolds with different biocompatible OEG linker structures. Our approach involves the construction of two building blocks: terminal propargylated OEG-tethered BAs and multiazide substituted per-*O*-acetylated CDs in the first stage, followed by a regioselective 1,3-dipolar cycloaddition reaction and a subsequent de-*O*-acetylation reaction to provide the desired multivalent BA-CD conjugates. This general strategy is particularly suitable for the rapid assembly of structurally well-defined multivalent compound libraries based on CD scaffolds. Due to the numerous free hydroxyl groups at the CD scaffold and the OEG linker, the generated BA-CD derivatives are expected to display high solubility and good compatibility in biological environments. No obvious cytotoxicity to MDCK cells was observed for these conjugates at concentrations up to 100 μM. Further in vitro testing showed that four conjugates, **51** and **69**–**71**, were potent against A/WSN/33 (H1N1) virus with IC_50_ values below 10 μM. The work presented herein demonstrated that multivalent BA derivatives have the potential to fight viral infection.

## Data Availability

The data of this study is contained within the article or Appendix A. The data presented in this study are available in this manuscript.

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
