# Peer review of "Facial Synthesis and Bioevaluation of Well-Defined OEGylated Betulinic Acid-Cyclodextrin Conjugates for Inhibition of Influenza Infection"

_molecules, 2022, doi:10.3390/molecules27041163_

Round 1
Reviewer 1 Report
This manuscript follows a series of studies on the anti-influenza effects of multivalent pentacyclic triterpenes from the authors, with synthesis and cellular assays of numerous multivalent compounds.
While the results are of good quality, the introduction and discussion can be improved.
First, references should be included for sentences ending at lines 40, 45, 56, 61, 83, and 246, respectively.
Second, more details should be provided for the general readership of Molecules. It is worth introducing that hexa-, hepta-, and octa-valent BA corresponds to alpha, beta, and gamma-cyclodextrin, respectively. Positions C-3 and C-28 of BA should be indicated graphically. The protein-like structure at top right corner of Figure 1 should be explained. Numbering of compounds in Scheme 3 should be more clear and it is not straightforward to see why compounds 51-68 only correspond to m = 0, 1, 2 and etc. “SI” should be spelled out in the footnote of Table 1 as done for IC50 and CC50.
Third, can the authors discuss that the stronger cytotoxicity of parent compound 31 may be due to its better cell permeability than its conjugates?
Fourth, the authors have previously employed surface plasmon resonance to study interactions between compound 51 and influenza HA protein, which should be noted in line 249-250. Can the authors discuss why they chose not to perform SPR assays for new conjugates described in this study? It appears that the authors have access to sufficient amount of HA protein and high-field (600 MHz) NMR spectrometer. So while not completely necessary, it would significantly add to the structural aspect of this study, and previous related studies, if the authors could perform saturation-transfer difference NMR assays of BA itself or its conjugates and even its/their competition binding with known binders of the HA protein.
Fifth, minor language issues at lines 909-911 should be corrected, and lines 114-116 should be removed. The “activity” at lines 95 and 110 should be “activities”.

Reviewer 2 Report
The manuscript entitled "Facial Synthesis and Bioevaluation of Well-Defined OEGylated Betulinic Acid Cyclodextrin Conjugates for Inhibition of Influenza Infection". Title, abstract and overall rationale of work to some extent is satisfactory. However, there are still some major concerns, which needs to be addressed and needs substantial revision.
1) Keywords: Do not repeat words from title and correct it and add suitable keywords
2) Introduction section is written well but author need to concise this section.
3) What about the cytotoxicity of BA-CD conjugates especially 51, 69, 70, 71, 75 and 78 compounds in MDCK cells. Author should be mention in the manuscript.
4) Another question: Author already know that compounds no. 58, 80 and 82 are toxic then why author further used for this study. However, author showing the compound no. 58 is showing good inhibitory activity but still we don’t know this come due to toxicity or really this compound work or inhibit the influenza virus activity.
5) Author must be check the cytotoxicity of other cell line such as fibroblast cells for comparable. Especially compounds no. 51, 69, 70, 71, 75 and 78
6) For results validity, author must be perform in-vivo activity and show the inhibition rate in the mice model.
7) Author need to explain why author did only this specific influenza strain (A/WSN/33) and why author is not use another strain like A/PR/8/34 virus to compare their study and show these all compounds inhibit all kind of influenza strain. I suggest author perform this experiment and clarified the compound activities.
8) There are lot of punctuation and typographical errors throughout in the manuscript. Unfortunately, I can’t correct throughout. It must be rechecked by native English speaker.
9) Author need to increase the resolution of figure 2.
Round 2
Reviewer 2 Report
I have completed my evaluation of your manuscript and I found, authors have addressed all the concerns raised in the previous version of the manuscript and the quality has improved after incorporating required modifications. Therefore, the manuscript may be considered for publication in this Journal.
